# Population genomics of ancient and modern *Trichuris trichiura*

Stephen R. Doyle [1,17 ✉], Martin Jensen Søe [2,17], Peter Nejsum[3], Martha Betson[4], Philip J. Cooper[5,6], Lifei Peng[7], Xing-Quan Zhu [8], Ana Sanchez [9], Gabriela Matamoros[10], Gustavo Adolfo Fontecha Sandoval [10], Cristina Cutillas[11], Louis-Albert Tchuem Tchuenté[12], Zeleke Mekonnen [13], Shaali M. Ame[14], Harriet Namwanje[15], Bruno Levecke[16], Matthew Berriman [1], Brian Lund Fredensborg [2] & Christian Moliin Outzen Kapel [2✉]

The neglected tropical disease trichuriasis is caused by the whipworm *Trichuris trichiura*, a soil-transmitted helminth that has infected humans for millennia. Today, *T. trichiura* infects as many as 500 million people, predominantly in communities with poor sanitary infrastructure enabling sustained faecal-oral transmission. Using whole-genome sequencing of geographically distributed worms collected from human and other primate hosts, together with ancient samples preserved in archaeologically-defined latrines and deposits dated up to one thousand years old, we present the first population genomics study of *T. trichiura*. We describe the continent-scale genetic structure between whipworms infecting humans and baboons relative to those infecting other primates. Admixture and population demographic analyses support a stepwise distribution of genetic variation that is highest in Uganda, consistent with an African origin and subsequent translocation with human migration. Finally, genome-wide analyses between human samples and between human and non-human primate samples reveal local regions of genetic differentiation between geographically distinct populations. These data provide insight into zoonotic reservoirs of human-infective *T. trichiura* and will support future efforts toward the implementation of genomic epidemiology of this globally important helminth.

[1] Wellcome Sanger Institute, Hinxton, Cambridgeshire, UK. [2] Department of Plant and Environmental Sciences, University of Copenhagen, Frederiksberg, Denmark. [3] Department of Clinical Medicine, Aarhus University, Aarhus N, Denmark. [4] School of Veterinary Medicine, University of Surrey, Guildford, UK. [5] Institute of Infection and Immunity, St George's University of London, London, UK. [6] School of Medicine, Universidad Internacional del Ecuador, Quito, Ecuador. [7] Department of Parasitology, School of Basic Medical Sciences, Guangdong Medical University, Zhanjiang, Guangdong Province, People's Republic of China. [8] College of Veterinary Medicine, Shanxi Agricultural University, Taigu, Shanxi Province, People's Republic of China. [9] Department of Health Sciences, Brock University, St. Catharines, Ontario, Canada. [10] Microbiology Research Institute, Ciudad Universitaria, Universidad Nacional Autónoma de Honduras, Tegucigalpa, Honduras. [11] Departamento de Microbiología y Parasitología, Facultad de Farmacia, Universidad de Sevilla, Sevilla, Spain. [12] Faculty of Sciences, University of Yaoundé I, Yaoundé, Cameroon. [13] Institute of Health, School of Medical Laboratory Sciences, Jimma University, Jimma, Ethiopia. [14] Public Health Laboratory Ivo de Carneri, Pemba, Tanzania. [15] Vector Control Division, Ministry of Health, Kampala, Uganda. [16] Department of Translational Physiology, Infectiology and Public Health, Ghent University, Ghent, Belgium. [17] These authors contributed equally: Stephen R. Doyle, Martin Jensen Søe. ✉email: stephen.doyle@sanger.ac.uk; chk@plen.ku.dk

The human-infective whipworm, *Trichuris trichiura*, is a soil-transmitted helminth (STH) responsible for trichuriasis, a neglected tropical disease (NTD) estimated to affect as many as 500 million people worldwide[1]. As an intestinal parasite, an infection begins by the ingestion of embryonated eggs in contaminated food or soil; eggs migrate to the large intestine and hatch, after which emerging larvae burrow and establish an intracellular niche within intestinal epithelia[2] where they develop into adult stages that can remain in situ for years[3]. Although low and moderate infection burdens typically remain asymptomatic, enabling efficient transmission in regions with poor sanitation such as endemic rural settings, chronic infections with high worm burdens cause a range of debilitating gastrointestinal symptoms and can lead to nutritional deficiencies and delays in physical and cognitive development, especially in children[4]. Infections with *T. trichiura* (and the other STHs, i.e. *Ascaris lumbricoides* and the hookworms *Ancylostoma duodenale*, *A. ceylanicum*, and *Necator americanus*) are primarily treated with benzimidazole anthelmintics (albendazole or mebendazole); however, the efficacy of these drugs for the treatment of *Trichuris* is suboptimal when given as a single dose[5,6] and few alternative drugs are available. In endemic regions, treatment is targeted towards pre-school and school-aged children, women of reproductive age, and high-risk working adults via annual or biannual mass drug administration (MDA) campaigns. In 2019, more than 613.1 million children, representing 60% of treatment coverage of children at-risk of STH infection, were treated with anthelmintics[7]. It is almost certain that this number will only increase in the near future, as STHs are targeted for elimination as a public health problem by 2030 in line with the WHO Sustainable Development Goals for NTDs[8] and the WHO road map for NTDs 2021–2030.

Humans have been parasitised by *T. trichiura* for millennia. Although now generally restricted to tropical and subtropical regions[1], this parasite was once a globally distributed pathogen. Parasite eggs have been found in human coprolites (fossilised faeces) from archaeological sites dated back to 7,100 BC[9–11], including locations in Europe and North America where autochthonous infections are now unusual[12–16]. However, whipworms are known to infect a broad range of mammals; over 70 species have been described within the genus *Trichuris*, and while generally host-specific, cross-host species transmission of individual parasite species has been reported, including between humans and pigs[17], humans and dogs[18], or humans and non-human primates[19–23]. Parasites that can interchangeably infect human and non-human hosts represent a credible challenge to control campaigns, as non-human hosts may act as a reservoir in which parasites can evade treatment and subsequently become a source of infections for treated populations. Understanding the historical and modern dispersal of *T. trichiura* in human and non-human hosts is likely required to achieve elimination. Further, the goal of eliminating STHs as a public health problem by MDA is threatened by the emergence of resistance to the benzimidazole compounds used to control them, as observed in other parasitic worms (in particular, veterinary helminths) frequently exposed to the drug[24]. Although evidence suggesting that resistance is emerging in *T. trichiura* is limited, understanding population connectivity within and between endemic regions will inform the likelihood and rate of resistance alleles spreading when they arise.

Here, we describe the first population genomic analysis of *T. trichiura*. Using whole-genome sequencing data of modern whipworms collected from human and non-human primate hosts, together with ancient samples preserved in archaeologically-defined latrines and deposits[13], we describe broad-and fine-scale genetic structure, admixture, and population demographics of geographically and genetically distinct parasite populations. We also explore genome-wide evidence of local adaptation between different human worms and between human and animal worms. These genome-wide variant data will support future efforts toward implementing genomic epidemiology of this globally important STH.

## Results

**Sequencing of ancient and modern *Trichuris trichiura*.** We have generated whole-genome sequencing data from 44 modern and 17 ancient samples (Supplementary Data 1), resulting in an average coverage of 9.31× and 0.66× of the nuclear genomes and 613× and 95× of the mitochondrial genomes, respectively (Supplementary Data 2). The samples analysed were derived from a broad geographic distribution and included 18 populations from nine countries spanning Africa, Central America, Asia, and Europe (Fig. 1a). The modern samples were composed of 37 worms obtained from human hosts, as well as seven samples obtained from captive animal hosts, including two parasites from baboons (*Papio hamadryas*), two from colobus monkeys (*Colobus guereza kikuyensis*) and three from leaf monkeys (*Trachypithecus francoisi*). All modern samples were single worms, except for samples from Cameroon ($n = 5$) and Tanzania ($n = 5$) which were pools of eggs.

The ancient samples ($n = 17$) derived from DNA extracted from pooled eggs were obtained from archaeological latrines and dig sites, primarily from Denmark (five locations) as well as from the Netherlands (two locations) and a single site in Lithuania, which individually have been dated to span the last thousand years (Supplementary Fig. 1). Thus, these samples represent the oldest helminth samples and likely the oldest eukaryotic pathogens[25] from which whole-genome sequencing data has been derived to date. Reads derived from ancient samples displayed increased sample damage due to deamination relative to the modern samples (Supplementary Fig. 2; Supplementary Data 2), particularly in the first two bases of each read, which were removed before downstream analyses.

Joint genotyping followed by stringent filtering identified 1,888 mitochondrial and 6,933,531 nuclear variants in the sample cohort. Considering the variation in depth of coverage and degree of variant missingness between samples, we separately analysed subsets of samples with either mitochondrial or nuclear sequence data using variants with at least 3x coverage across at least 80% of sites within each dataset. The nuclear variants were split further into sex-linked and autosomal datasets. This approach limited the impact of biases as a result of genetic and sequencing heterogeneity; although a negative correlation between coverage and heterozygosity was observed in the extremely low coverage ancient samples (as a result, 15 of 17 samples were excluded from most downstream analyses of nuclear genetic variation due to low coverage), there was no significant impact of coverage variation on genetic diversity in any other population analysed (Supplementary Fig. 3). Depth of coverage analyses comparing autosomal and sex-linked scaffolds revealed 17 male (XY; expected 0.5× depth of sex-linked scaffolds relative to autosomes) and 17 female (XX; expected 1:1 ratio of sex-to-autosomal depth) worms in the individual worm sequencing data; the datasets derived from pooled eggs (ancient samples, and modern samples from Cameroon & Tanzania) revealed intermediate coverage due to the presence of mixed-sex eggs within the pools (Supplementary Fig. 4; Supplementary Data 2). The colobus and leaf monkey samples were also excluded depending on the analysis.

**Broad-scale genetic diversity of ancient and modern helminths.** To investigate broad-scale genetic diversity within the global cohort, we first performed a principal component analysis of

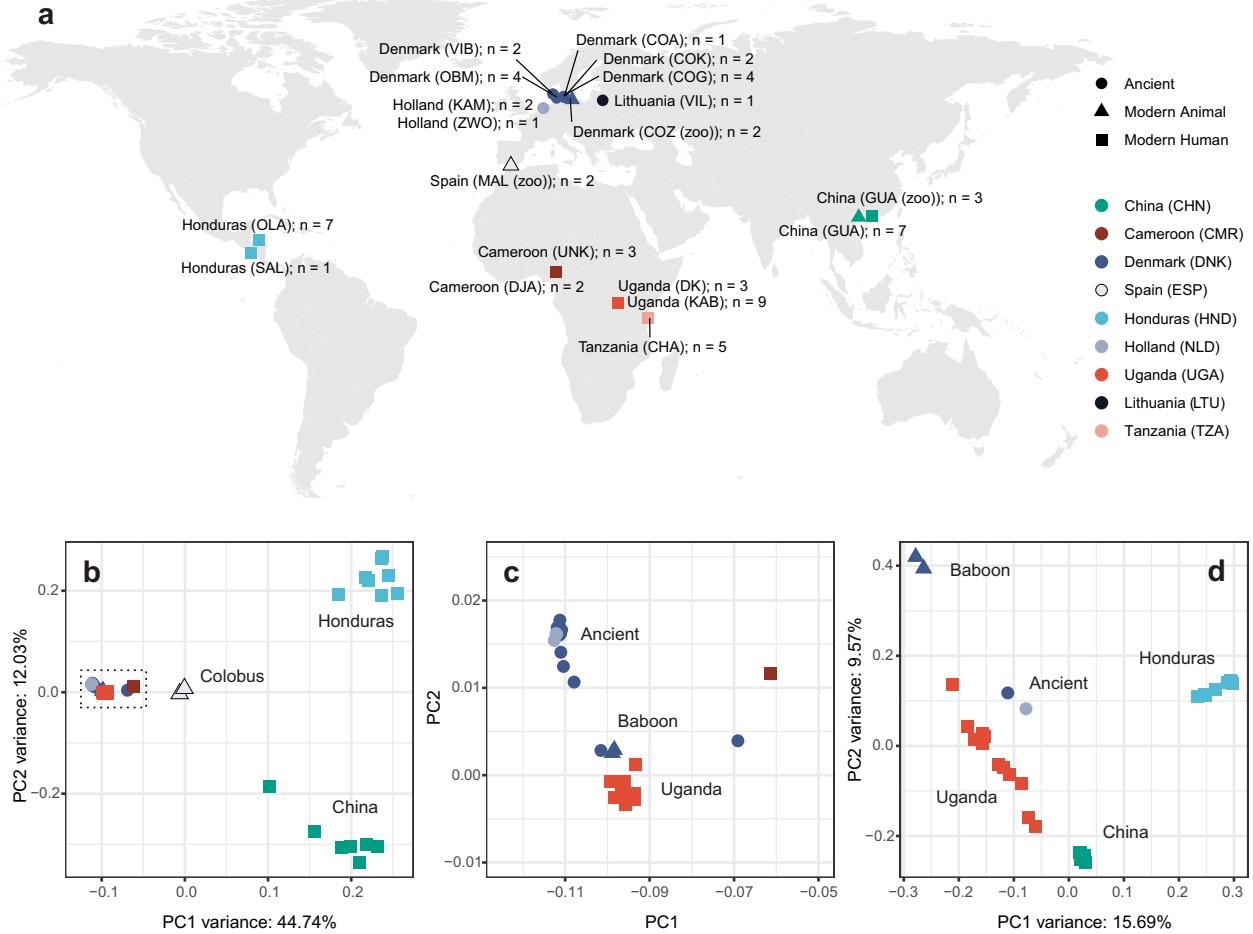

**Fig. 1 Global sampling distribution and broad-scale genetic relatedness of ancient and modern *Trichuris trichiura*. a** World map showing the approximate sampling locations of samples used in the study, highlighting geographic regions of ancient and modern sampling and host species. Sample site abbreviations: CHA = Chake; COA = Adelgade, Copenhagen; COG = Gammel Strand, Copenhagen; COK = Kultorvet, Copenhagen; COZ = Copenhagen Zoo; DJA = Dja Fauna Reserve; GUA = Guangdong; KAB = Kabale, Nyakitokoli; KAM = Kampen; MAL = Málaga Zoo; OMB = Odense By Midte; OLA = Olanchito; SAL = San Lorenzo; VIB = Viborg; VIL = Vilnius; ZWO = Zwolle. **b** Principal component analysis of mitochondrial diversity (56 samples, 802 variants). **c** Zoomed-in view of the cluster of samples indicated by the dashed box in **b**, containing ancient, Ugandan, and baboon-derived samples. **d** Principal component analysis of nuclear diversity using a subset of higher quality samples (31 samples, 2,544,110 autosomal variants).

mitochondrial variation (Fig. 1b). Three genetically defined clusters were identified, clearly separating samples from China and Honduras from the third cluster containing samples of mixed origin. Closer inspection of this third cluster identified closely related but genetically distinguishable ancient, modern Ugandan and baboon samples (Fig. 1c). To provide further granularity, we analysed autosomal genetic variation with a subset of higher quality samples, which revealed greater levels of genetic diversity within the Ugandan samples, and provided more apparent differentiation between geographically distinct human worms, as well as between the ancient and baboon samples, than the mitochondrial DNA alone (Fig. 1d). We also sampled additional populations from Africa (Cameroon & Tanzania); however, there was limited genetic information due to the very low sequencing coverage obtained, likely due to the fact we sampled only unembryonated eggs (with environmentally robust eggs shells, and thus, difficult to crack) rather than adult worms and, therefore, the DNA concentration was very low and suboptimal for sequencing[26]. Further sampling would be needed to precisely place the samples from these regions relative to the larger and better genetically defined cohort.

Whipworms obtained from leaf and colobus monkeys were highly genetically distinct from the human/baboon group and

each other (Supplementary Fig. 5a). To better understand their phylogenetic placement in the *Trichuris* genus, we assembled mitochondrial genomes from all modern worm samples using an iterative baiting and mapping approach and compared them to publicly available whole mitochondrial genomes. Whipworms from leaf monkeys formed a sister group to the human and non-human primate clades, whereas those from the colobus group were more closely aligned with pig whipworm *T. suis* samples in a separate cluster (Supplementary Fig. 5b). This finding further supports that these animal-derived *Trichuris* spp. are genetically distinct species from the human-and baboon-infective *T. trichiura*.

**Patterns of admixture between populations across space and time**. The generation of sequencing data from 17 ancient samples provided a unique opportunity to understand patterns of parasite diversity in Northern Europe over time. Measures of genetic differentiation between populations revealed some degree of isolation-by-distance, where geographically close populations were genetically indistinguishable, for example, COG and COA in Denmark ($F_{ST} = 0.009$) and KAM and ZWO in Holland ($F_{ST} = 0.004$), but more distantly located populations are more

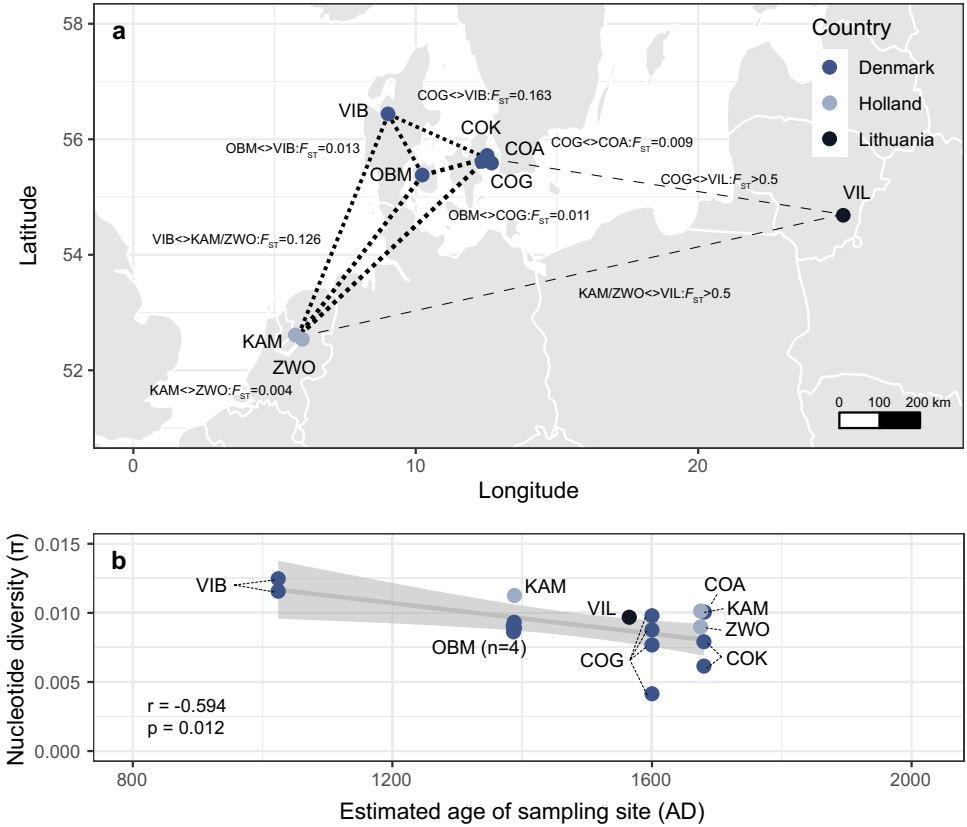

**Fig. 2 Changes in ancient parasite diversity across space and time. a** Genetic differentiation between ancient sample sites based on pairwise estimates of mitochondrial genome diversity (pooled sample $F_{ST}$) is shown. The width of the dashed line connecting pairs of sample sites reflects the degree of similarity between mitochondrial genomes. **b** Comparison of the levels of nucleotide diversity ($\pi$) within the ancient mitochondrial genomes over time. Pearson's correlation (r) and associated p-value (cor.test; two-sided; adjusted p-value ["holm"]), together with the linear regression (grey trendline line) and 95% confidence level interval (grey shaded area), are shown. Samples are coloured based on the scheme in Fig. 1. Sample site abbreviations: COA = Adelgade, Copenhagen; COG = Gammel Strand, Copenhagen; COK = Kultorvet, Copenhagen; OMB = Odense By Midte; VIB = Viborg; KAM = Kampen; ZWO = Zwolle; VIL = Vilnius.

genetically distinct (Fig. 2a). Two exceptions were found. First, VIB in Denmark was more genetically distinct than expected to other nearby populations based on location; however, these were also the oldest samples (dated 1018–1030 AD and ~350 years older than the next sampled population) and, therefore, the differentiation is likely exacerbated by genetic change over time. Second, although geographically distinct, samples from VIL in Lithuania were highly genetically differentiated ($F_{ST} > 0.5$), suggesting little, if any, genetic connectivity to the populations in Denmark and Holland. However, this finding is consistent with the human genetic ancestry of Danish Vikings that predominantly travelled eastward, unlike Swedish Vikings that travelled more extensively into the Baltic region[27]. To understand the impact of time on these populations, we compared nucleotide diversity against the estimated age of the samples, revealing a small but significant loss of variation over time towards the present (Fig. 2b; $r = -0.59$, $p = 0.012$). The oldest and genetically distinct VIB samples were also among the most diverse; this relationship between diversity and time was robust to DNA damage artefacts (deamination susceptible C-to-T and G-to-A substitution variants removed; $r = -0.49$, $p = 0.047$), however, the trend was still negative but less pronounced and not significant when the VIB samples were removed ($r = -0.28$, $p = 0.305$). We note that this decline was well before modern interventions were used to control parasites such as *T. trichiura* and that poor sanitary conditions should have favoured transmission. However, the lower temperatures at northern latitudes

would have significantly reduced eggs' embryonation rate and viability in the environment, allowing for a maximum of one parasite generation per year. As a result, parasite population sizes may have undergone more significant fluctuations in such suboptimal conditions, including bottlenecks that would have reduced diversity relative to modern populations currently endemic.

To more formally describe the genetic relationships between ancient, modern and baboon populations, we used NGSadmix to visualise the ancestral composition of samples. We estimated three ancestry components (K = 3) (Fig. 3a), broadly consistent with the global population structure in mitochondrial and nuclear PCAs (Fig. 1b–d); ancient and baboon samples formed a cluster that was distinct from each of China and the Honduras populations. Ugandan samples showed mixed ancestry with China and ancient/baboon samples. To explore this further, we determined admixture proportions across a range of K values (K = 2–10) (Supplementary Fig. 6). Subtle evidence of shared ancestry between China and Honduras was observed, but this disappears at K ≥ 3. Ugandan samples displayed the most diverse ancestry profile, with complex mixtures of private components not found elsewhere; we note that the first two Ugandan samples of the admixture plot that maintain the same admixture profile throughout were sampled from the same host and are likely 2nd degree relatives (MN_UGA_DK_HS_001 & MN_U-GA_DK_HS_002; kinship coefficient [Θ] = 0.34; Supplementary Fig. 7). Similarly, samples from Honduras also increased in

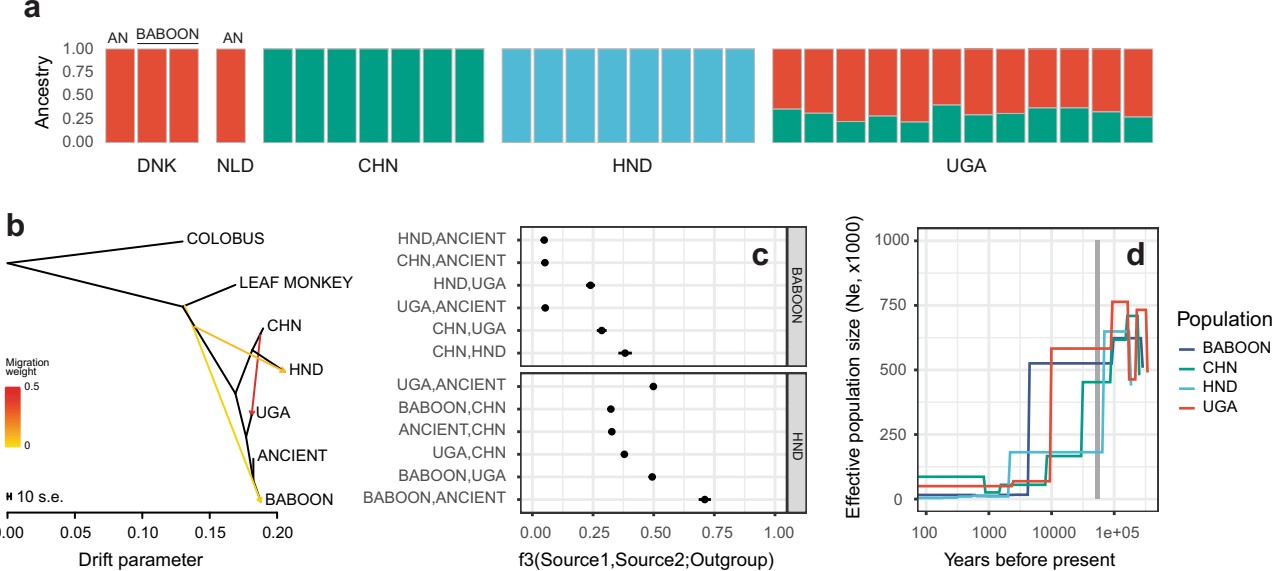

**Fig. 3 Fine-scale genetic relationships and admixture between global populations. a** Admixture plot depicting population ancestry proportions determined using NGSadmix (K = 3, variants = 484,914). See Supplementary Fig. 6 for a complete analysis of K from 2 to 10. Ancient (AN) and baboon samples are highlighted. **b** Treemix maximum likelihood tree of ancient and modern samples, including Colobus and Leaf monkey samples as outgroups, showing three migration edges. See Supplementary Fig. 8 for a complete analysis of migration edges from 0 to 5. **c** Outgroup $f3$ statistics were determined using $qp3Pop$ in ADMIXTOOLS to compare allele frequency correlations between two source populations (indicated on the left side of the panel) relative to an outgroup population (right side of the panel). $f3$ values ($f3 = [f_{outgroup}-f_{source1}]*[f_{outgroup}-f_{source2}]$, where $f$ is the allele frequency, averaged over all variable sites) are indicated by a point and whiskers represent the standard error, calculated using a weighted block jackknife (2,575,411 SNPs; $n$ = 3 blocks; default parameters). For all data presented, the Z score was significant ($|Z| > 3$). **d** The population demographic history of each population was determined using SMC++ to compare effective population size (Ne) over recent evolutionary history. The grey vertical box highlights the period between 50 and 60 thousand years ago, coinciding with the migration of modern humans out of Africa. Sample abbreviations: CHN = China; UGA = Uganda; HND = Honduras.

diverse components throughout the range. Interestingly, the ancient and baboon samples maintained a single fixed ancestry profile throughout all values of K tested.

To quantify genetic admixture, we used Treemix to estimate migration between nodes under a maximum likelihood framework (Fig. 3b). Three migration edges were supported (see Supplementary Fig. 8 for trees and associated residual heatmaps for edges = 0–5) and emphasised admixture between Ugandan and Chinese populations. Interestingly, there was evidence implicating a lower level of admixture between the base of the tree adjacent to the leaf monkey branch and both Honduran and Baboon samples. While this does not directly implicate hybridisation between non-human-infective and human-infective whipworms specifically, hybridisation between the phylogenetically related but pig-infective *T. suis* and *T. trichiura* has been described previously in Ecuador[28]. We further quantified these admixture results using *f*-statistics on the nuclear variants (Fig. 3c). Using baboon samples as an outgroup, we further identified the relationship between China and the Honduras populations and found that this relationship was closer than that between Ugandan and Honduras populations. Investigations of the mitochondrial markers *rrn*L and *nad*1 from humans in Uganda, China and Ecuador (geographically close to Honduras) identified genetic similarities between parasites from China and Ecuador, which were distinctly different from those from Uganda[17]. Using baboon as an outgroup reduced our ability to measure more subtle differences between the closely related baboon, ancient, and Ugandan populations; to address this, we used Honduras as an outgroup, which provided greater resolution to show the closer relationship between the baboon and ancient samples, relative to comparison with the Ugandan samples.

Finally, we used population demographic analyses to characterise the effective population size of the parasite populations over time (Fig. 3d), revealing that current populations are a fraction of the size of the historical populations from which they were derived. All populations showed a substantial decline in effective population size between 50 and 100 thousand years ago; Ugandan and baboon populations did maintain a significantly higher population size until approximately ten thousand years ago, after which they underwent further decline. Together with admixture analyses, these findings allow us to hypothesise about the potential dispersal of *T. trichiura*. Although we have made specific assumptions about important but unmeasurable parameters of the population demographic model — specifically, mutation rate and generation time — and therefore, the precision of these estimates is low, these data are consistent with bottlenecks in parasite populations that likely occurred as modern humans migrated out of Africa approximately 50–60 thousand years ago[29]. The intermediate *f3* statistics between Uganda and China suggest a stepwise migration pattern, first from Africa to Asia and then from Asia to the Americas. This pattern would be consistent with human migration, the latter of which perhaps via a crossing of the Beringia land bridge that connected the two continents until approximately 11 thousand years before the present. This migration was distinct from that established in Europe, indicated by the lower degree of shared variation between ancient and Chinese samples but almost no sharing between the ancient and the Honduran samples. It has been argued that some intestinal parasites may not have survived the Beringia crossing due to unfavourable environmental conditions to sustain their lifecycle, and thus, the genetic connectivity between parasite populations may reflect additional migration to the Americas

through trans-pacific or coastal routes[9]. To explore this here, we compared the proportion of variation shared between Ugandan, Chinese and Honduran samples. While a large proportion of variation was private (in total, 46.6% of variants are present only in one of the three populations) or shared by all three populations (26.8%), 3.2% of variants are shared between the Ugandan and Honduran populations that are absent from Chinese samples (Supplementary Fig. 9); while tenuous due to the low sample numbers and sparse geographic sampling, such variation would support a low level but independent introduction of parasites to the Americas from Africa.

**Genome-wide patterns of selection and adaptation.** Distinct geographic clustering of Ugandan, Chinese, and Honduran populations prompted us to examine evidence of diversifying selection that might indicate local adaptation within the human-infective parasites. Nucleotide diversity ($\pi$) varied between populations, with the highest diversity observed in Uganda (genome-wide median $\pi = 0.0120$), followed by China ($\pi = 0.0103$), baboon ($\pi = 0.0092$), and then Honduras ($\pi = 0.0079$), consistent with a global dispersal of *T. trichiura* originating in Africa (Supplementary Fig. 10). Ancient samples presented the lowest diversity of all populations ($\pi = 0.0037$). There was variation throughout the genomes in each population, the most notable of which was the difference in sex-linked to autosomal diversity ($\pi_{X\text{-linked}}/\pi_{autosome} = 0.72$), which was consistent with the diversity expected ($\pi_{X\text{-linked}}/\pi_{autosome} = 0.75$) for an organism with a $XX_{female}/XY_{male}$ sex chromosome system. A genome-wide analysis of nucleotide divergence (Dxy) between populations revealed a strong correlation with nucleotide diversity within populations (Supplementary Fig. 11). However, divergence calculated using a sliding window of $F_{ST}$ between pairs of populations identified local regions of differentiation that may indicate local adaptation (Fig. 4a). For each pairwise comparison (UGA vs CHN: Weir and Cockerham weighted $F_{ST} = 0.116$; UGA vs HND: $F_{ST} = 0.279$; CHN vs HND: $F_{ST} = 0.269$), we identified 373, 358, and 393 genes in region of high divergence (defined as the top 5%), respectively. Analysis of gene ontology (GO) terms (Supplementary Data 3, 4, and 5 for each pairwise comparison, respectively) did not, however, identify any enriched terms in the UGA vs CHN and CHN vs HND genesets that would have suggested the selection of multiple genes of common function. However, the UGA vs HND genes were enriched in the molecular function GOterms anion:cation symporter activity (GO:0015296; $p = 3.988e\text{-}2$) and cation:chloride symporter activity (GO:0015377; $p = 3.988e\text{-}2$). We extended this analysis to compare Ugandan and baboon samples, given their close genetic relationship despite being isolated from different host species. Genome-wide mean $F_{ST}$ was intermediate ($F_{ST} = 0.148$) to that of the human-specific analyses, re-emphasising that despite divergence, baboon-infective whipworms are within the species-level of diversity expected of human-infective *T. trichiura*. Genome-wide, discrete regions of differentiation were observed, particularly on the sex-linked scaffolds (Fig. 4b; LG1 [X]). Regions of high divergence (top 5%) contained 339 genes (Supplementary Data 6) that were enriched for cellular-component GOterms, including NURF complex (GO:0016589; $p = 8.432e\text{-}5$), ISWI-type complex (GO:0031010; $p = 4.111e\text{-}4$) and chromatin (GO:0000785; $p = 1.544e\text{-}3$); these terms are associated with chromatin remodelling and transcriptional regulation of developmental genes[30] and may reflect selection on genes that require differential regulation during development between the different host species.

Anthelmintic resistance could have catastrophic consequences if it emerged in human infectious parasites. It was, therefore,

important to use this global collection to explore variation in and around $\beta$-tubulin, a gene long associated with variation in response to benzimidazole-class anthelmintics such as those used to treat trichuriasis. We identified nine polymorphic sites in the sex-linked $\beta$-tubulin gene (Gene id: *TTRE_0000877201*; Location: Trichuris_trichiura_1_001: 10684531-10686350) segregating within modern human populations; however, no variation within the codon positions P167, P198, or P200 that are typically associated with benzimidazole resistance was observed (Supplementary Fig. 12a), consistent with previous studies[23,31–33]. Further, there was little evidence of broader-scale genetic change on standing genetic variation in the region surrounding the $\beta$-tubulin gene that might be associated with positive selection on a gene within that region (Supplementary Fig. 12b, c).

**Discussion**

The whipworm *T. trichiura* is one of humans' most prevalent and globally distributed helminth pathogens. Here, we have undertaken a broad survey of genome-wide genetic diversity of *Trichuris* collected from human and non-human primates, including, to our knowledge, the oldest eukaryotic pathogens analysed using whole-genome sequencing to date. *Trichuris* spp., as a soil-transmitted helminth, completes part of its life cycle in the environment; unembryonated eggs are deposited in faeces where they undergo embryonation over two-to-four weeks under ideal, typically warm and moist conditions commonly found in tropical and subtropical regions. The remarkable resilience of eggs in the environment allows them to remain viable in the soil for at least a decade[34] and morphologically identifiable to genus level after thousands of years[9–11]. Here we demonstrate that DNA within eggs remains sufficiently preserved for whole-genome sequencing from samples up to one thousand years old, providing a unique and information-rich insight into parasites of the human past.

Analyses of the genomic diversity of ancient parasites and their genetic connectivity to modern parasite populations unexpectedly revealed a close genetic relationship to Ugandan and Baboon samples. Unfortunately, the ancient samples consisted of pools of eggs rather than individual worms, limiting investigation of population demographics and estimated split times between ancient and these closely related modern populations. Epidemiological estimates of *T. trichiura* prevalence have been proposed to be comparable between medieval populations in Europe (dated between 680 and 1700 CE) and modern, non-European endemic regions[16], and in Denmark, *Trichuris* infections were prevalent in children, particularly from rural areas up to at least the 1930s[35]. In contrast, the low genome-wide genetic diversity and declining nucleotide diversity over time of our ancient samples are consistent with a constrained and likely contracting population size. Whether this reflected actual population decline when the eggs were deposited, non-random sampling of genetically related parasite eggs within a defined archaeological site or non-random distribution of whipworm prevalence in the host population is unclear. However, the population demographics of these ancient northern European populations are likely different relative to populations found in more tropical conditions where it is currently endemic. The parasite development rate in the environment is temperature-dependent and decreases in cooler temperatures[36], and below 15 °C, development arrests. This is particularly relevant for parasite populations such as those once endemic in northern Europe, as the proportion of the year during which favourable conditions for parasite development (and thereby transmission) decreases at increasing latitudes. Thus, suboptimal environmental conditions necessary for parasite development and infection may partly explain regional differences in parasite prevalence, including local regions of population

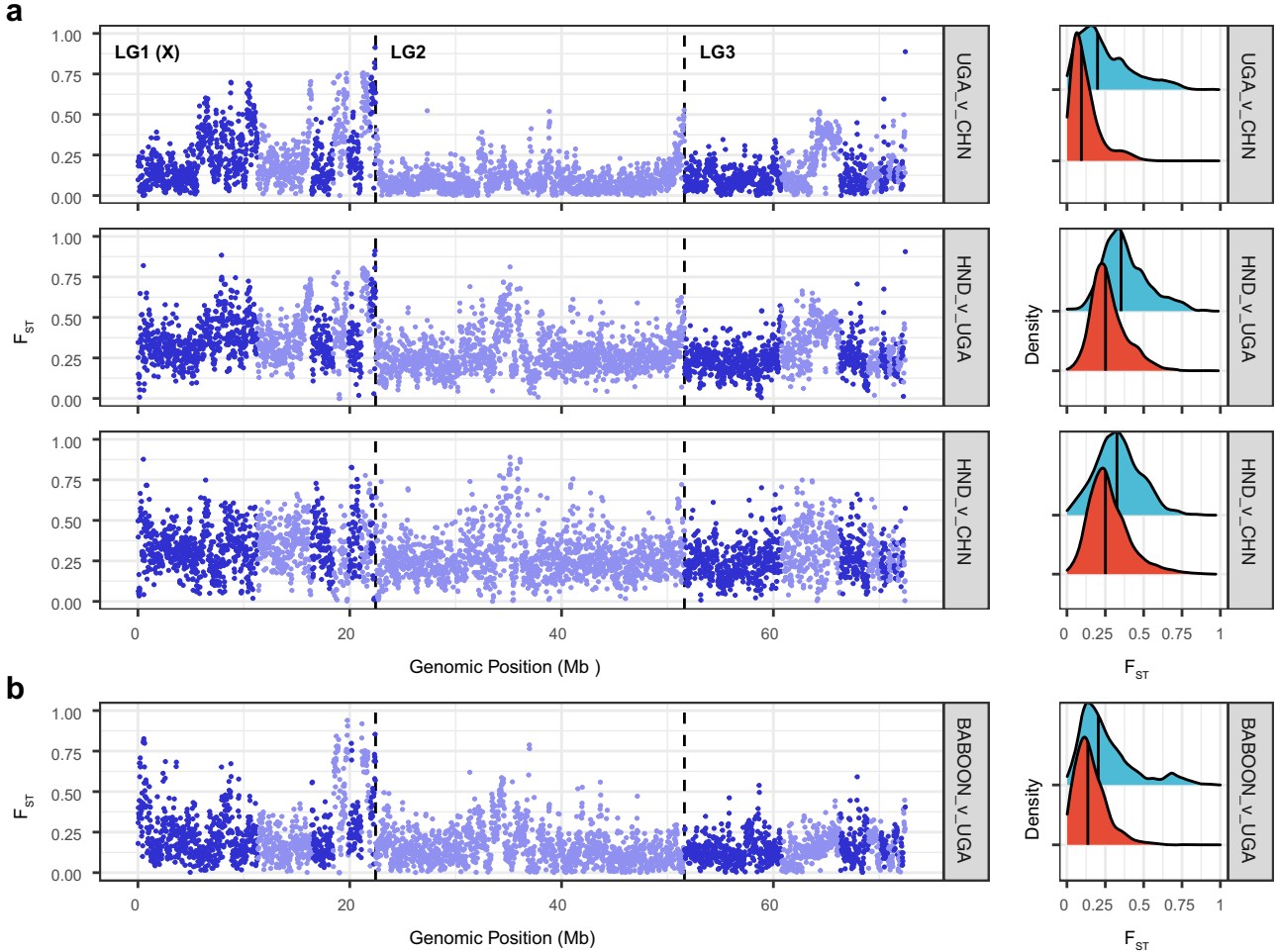

**Fig. 4 Genome-wide comparison of genetic variation between populations. a** Comparison of genome-wide genetic differentiation between human-infective *T. trichiura* from genetically and geographically defined populations. Pairwise $F_{ST}$ was measured in 20 kb windows between China and Uganda (top), Uganda and Honduras (middle), and China and Honduras (bottom). Alternating dark and light blue colours represent different scaffolds, with the vertical dashed lines separating the three chromosomal linkage groups (LG). The sex-linked linkage group is indicated as LG1 (X). **b** Comparison of closely related human-infective Ugandan and baboon-infective *T. trichiura*. Sample abbreviations: CHN = China; UGA = Uganda; HND = Honduras. Density plots in a and b show the distribution of $F_{ST}$ values for the sex-linked (blue) and autosomal (red) scaffolds. The median $F_{ST}$ value for each is shown (solid vertical line).

decline and diversity. Genomic analysis of ancient parasites from southern Europe and Africa will likely shed light on the missing links connecting the northern European and African parasites analysed here and the relationship between genetic diversity and endemicity in response to a changing climate.

Our genomic analysis of modern human and animal parasites further supports that *T. trichiura* is specific to humans, baboons, and maybe some other non-sampled primates and emphasises that parasite demography will depend on its host. Population demographic analyses of human infective whipworms support the hypothesis that global parasite populations suffered severe population bottlenecks coinciding with human migration out of Africa and that differential proportions of genetic admixture between these populations provide a plausible hypothesis by which parasites continued to infect modern humans as they distributed throughout the world. We note that one potential source of independent introduction of *T. trichiura* from Africa to the Americas resulting in shared parasite ancestry would likely be associated with the mass forced migration of Africans during the trans-Atlantic slave trade. We hypothesise that this genetic signature would be greater between West African (for which we have no samples) and American parasites than between Central

or East African parasites as sampled here and that West African parasites would be more suitable to test the degree of genetic admixture and thus impact of different migration events on the genetic diversity of endemic American parasites present today. Collectively, our data reinforce the important role that human migration has played in the spread of modern helminths throughout the world[9,17,37–39].

Comparative analysis of human and animal infective whipworms demonstrate that worms isolated from baboons likely represent a zoonotic reservoir of human-infective parasites in Africa. Unlike many zoonotic transmissions that involve a vector species, the likelihood of cross-species zoonotic transmission of a soil-transmitted pathogen like *T. trichiura* will be influenced by the degree to which humans and animals occupy and directly share ecological niches, for example, the ground or water supply. Baboons are large ground-dwelling primates that can be found close to human activity and likely share a common source of infection of STHs such as *T. trichiura*. This is in contrast to smaller, tree-dwelling non-human primates, including leaf and colobus monkeys, that would have less interaction with humans. However, genome-wide analyses between human and baboon parasites reveal discrete regions of the genome that differ between

hosts, potentially indicative of some degree of host adaptation. We note that whipworms were collected from only three species of captive non-human primates, and we lack sampling data from other apes that live in proximity to humans, particularly in natural settings. Further sampling of human- and a broader range of animal-infective parasites, particularly where cohabitation exists, combined with genome-wide analyses of selection, should provide further insight into the mechanisms by which host-specificity is maintained and cross-species transmission is tolerated.

Large-scale treatment of parasites using benzimidazole anthelmintics (as used by MDA programmes against STH, including *T. trichiura*) is known to select genetic variation in β-tubulin conferring resistance to treatment. This is particularly evident in veterinary helminths[37,40], where selection for benzimidazole resistance has rendered benzimidazole derivatives almost ineffective in some regions of the world, particularly against helminths of horses and sheep[24]. Emerging evidence suggests that MDA may be exerting similar selection pressure on STHs[41–43]. We found no evidence for selection within or surrounding the β-tubulin gene. However, we do not have any data on their phenotypic response or exposure history to anthelmintic treatment, and thus, we cannot determine whether the absence of known benzimidazole-resistance associated variants in our data indicates that: (i) all populations are phenotypically susceptible to anthelmintic treatment, or (ii) populations may or may not be phenotypically susceptible to benzimidazole treatment, but that selection on variation elsewhere in the genome, and not on variation in the β-tubulin gene, is responsible for drug response in *T. trichiura*. Future studies exploiting genome-wide analyses on parasite populations that are phenotypically well-defined for drug response[44] are needed to define the genetic architecture of drug response in *T. trichiura*. It will be particularly important to disentangle known variation in the tolerance to benzimidazole drugs by *T. trichiura*[45,46] from the emergence of loss of efficacy due to high drug exposure[6], especially if the promise of using molecular diagnostics to monitor the emergence of anthelmintic resistance is to be realised[47–50].

Our data establish the genetic framework for the genomic epidemiology of *T. trichiura*. It is almost certain that genomic surveillance will become an important tool for informing helminth control campaigns in the future[51,52]. Such data could be used to interpret variation between and changes within parasite populations, for example, to characterise population decline due to effective control or distinguishing between transmission, recrudescence, and loss of efficacy of the drugs used to control them. These require a comprehensive understanding of the underlying genetic diversity of the parasite throughout its range. Our data provide a significant first step towards this goal and will be enriched by further sampling throughout endemic regions to characterise finer-scale genetic connectivity and effective parasite transmission zones.

## Methods

### Sampling of ancient and modern *Trichuris* spp. from humans and animals.
Sampling details specific to each group or population of parasites are outlined below. As the focus of the study is on the genetics of whipworms, no patient-specific data or identifiers were included in any analyses performed. In all cases where parasites were collected from human patients, informed written consent was obtained from all participants or their guardians (in cases where the participant was a child) after being informed about the study in English and/or the local language. Samples were collected in line with ethical considerations and clearances as follows:

*Ancient*. *Trichuris* sp. eggs were isolated from ancient latrine samples in which a metagenomic study has previously identified DNA from *T. trichiura*[13]. The eggs were isolated from ancient human latrines or deposit sites in Denmark (1000–1700 AD), the Netherlands (1350–1850 AD), and Lithuania (1550–1580 AD) (Supplementary Fig. 1).

*China*. *T. trichiura* samples, previously defined as genetically distinct from pig-derived *Trichuris* (i.e. *T. suis*) by mitochondrial genome and nuclear genetic markers[53], were collected directly from the caecum of a human patient during surgery in Zhanjiang People's Hospital in Zhanjiang, Guangdong Province, China (CHN_GUA). These worms were sent to the Department of Parasitology, Guangdong Medical University, for identification.

*Colobus guereza kikuyensis*. Worms, previously defined as *Trichuris colobae n. Sp.*[54], were collected from the caecum during necropsy of *Colobus guereza kikuyensis* (Mantled Guereza or Eastern Black-and-White Colobus) (ESP_MAL [zoo]), which died of natural causes at the Fuengirola Zoo (Málaga, Spain).

*Papio hamadryas*. Worms from baboons were recovered from the caecum during routine post mortem examination of animals culled for management reasons at Copenhagen Zoo, Denmark[23].

*Honduras*. Samples were collected as a joint effort by Ana Sanchez (Brock University, Canada), Gustavo A. Fontecha and Gabriela Matamoros (Universidad Nacional Autónoma de Honduras (UNAH), Honduras) (HND_OLA and HND_SAL). Sample collection protocols were reviewed and approved by the Brock University Bioscience Research Ethics Board and Comité de Ética de Investigación, Maestría en Enfermedades Infecciosas y Zoonóticas, Facultad de Ciencias, UNAH.

*Trachypithecus francoisi*. *Trichuris* spp. samples were collected during necropsy from the caecum of a captive *Trachypithecus francoisi* (François Leaf-Monkey), which was humanely euthanised due to acute gastric dilation (People's Republic of China) (CHN_GUA [zoo])[55].

*Uganda*. Worms were recovered after anthelmintic treatment from the faeces of children in Uganda (UGA_KAB) as previously described[56]. Permission was obtained from the Ministry of Health and the National Council of Science and Technology in Uganda, and the Danish Central Medical Ethics Committee recommended the study. A subset of worms was recovered after anthelmintic treatment from a Danish patient infected with *T. trichiura* from Uganda (UGA_DNK) as previously described[3].

*Cameroon*. Stool samples were collected with a focus on monitoring the efficacy of mebendazole for the treatment of soil-transmitted helminths[6]. Permission was obtained from the National Ethics Committee in Yaoundé, Cameroon (Sep 2011).

*Tanzania*. Stool samples were collected as previously described[6]. Permission was obtained from the Ministry of Health, Zanzibar Revolutionary Government, Tanzania (July 2012).

Individual worms were washed extensively in tap water or saline solution and preserved in 70% ethanol. When possible, the sex of individual worms was determined through microscopic examination before DNA extraction.

**DNA extraction and high-throughput sequencing.** Ancient samples were processed for egg extraction in a dedicated paleoparasitological laboratory at the Department of Plant and Environmental Sciences (PLEN), University of Copenhagen. Helminth eggs were concentrated by selecting particles based on flotation in high-density liquid sedimentation and size filtration. Briefly, 10–503 g from each sample was manually homogenised using a sterile scalpel blade before separating into 50 mL tubes, each containing ~10–15 g. TE buffer (Tris-HCl 10 mM, EDTA 1 mM) was added to each tube to resuspend and then incubated for 3–6 days with daily mixing. Samples were pelleted (3000 RPM for 5 min), the supernatant was removed, and the pellet was resuspended in flotation buffer (FB) (glucose monohydrate 375 g/L+ sodium chloride 250 g/L). Samples were centrifuged (400–1000 RPM for 5 min), and the supernatant was sieved on stacked, disposable, autoclaved, 100 μm and 22.4 μm nylon filters. All tubes from a particular sample were processed on the same set of filters. Flotation, centrifugation and sieving were repeated for the pellet, and the filters were washed in demineralised water. The material was collected from the 22.4 μm, suspended again in FB buffer, centrifuged (400–1000 RPM for 5 min) and sieved on a new set of stacked 35.5 μm and 22.4 μm filters. Filters were washed in demineralised water, and sediments were collected from the 35.5 μm filter (most often termed 'sample ID' followed by 'A') and the 22.4 μm filter (most often termed 'sample ID' followed by 'B'). Sample material was washed three times in molecular grade water; 10% was then used for morphological examination, and the remaining 90% for DNA extraction. Morphological examination of eggs was performed on microscope slides at 100× and 400× resolution, and quantification of eggs was performed either on fixed microscope slides or McMaster counting chambers. The egg samples were then processed for DNA extraction in dedicated aDNA laboratories at the Centre for Geogenetics (CGG), University of Copenhagen, following strict aDNA-specific requirements. DNA extraction was performed using the PowerLyzer PowerSoil DNA isolation kit (MO BIO Laboratories, Cat: 12855-50) with minor modifications, including bead beating performed using Lysing Matrix I beads (MP Biomedicals, Santa Ana, California) for 90 s at 6.5× on the FastPrep-24 (MP Biomedicals) and DNA was eluted in 60 μl elution buffer. The complete protocol has been described previously[13].

DNA from modern worm samples were prepared as follows: (i) for samples collected from stool following treatment with anthelmintics, directly from a human patient via colonoscopy, and from baboon, DNA was extracted from whole adult worms using the MasterPure™ DNA Purification kit (Epicentre Biotechnologies, Cat: MC85200) at the Department of Veterinary Disease Biology (VDB), UCPH following the manufacturer's protocol with the following exception: worms were first homogenised in 300 μL lysis solution using a matching disposable plastic pestle and then incubated at 56 °C for three hours; (ii) for the colobus samples, DNA extraction from whole adult worms was performed at the Departamento de Microbiología y Parasitología, Universidad de Sevilla, Sevilla, Spain, using a DNeasy Blood & Tissue Kit (Qiagen, Cat: 69504) according to the manufacturer's protocol; (iii) for the leaf monkey samples, DNA extraction was performed at the Department of Parasitology, Lanzhou Veterinary Research Institute, Lanzhou, China as previously described[55]. DNA (400 ng) was fragmented to approximately 300–700 bp using a Diagenode BioRuptor using 2 or 4 cycles of 15 s ON/90 s OFF at instrument setting HIGH. Fragmented DNA was concentrated using a MinElute Reaction Cleanup Kit (Qiagen; Cat: 28206), after which the DNA concentration (Thermo Fisher Qubit dsDNA HS Assay Kit, Cat: Q32851) and library size (Agilent Bioanalyzer High Sensitivity DNA Kit; Cat: 5067-4626) was determined.

DNA sequencing libraries were prepared for all sample types, modern and ancient DNA extracts, using the NEBNext DNA Sample Prep Master Mix Set for 454 (New England Biolabs, Cat: E6070) kit with a modified protocol previously described for ancient samples[13]. DNA libraries were amplified in a single PCR reaction of 50 μL using: half of the prepared DNA library (12.5 μL), 5 U AmpliTaq Gold (Life Technologies, Cat: N8080241), 1× buffer Gold, 2 mM MgCl₂, 0.25 mM dNTPs, and 0.2 mM of primers (PE1.0 [AATGATACGGCGACCACCGAGAT CTACACTCTTTCCCTACACGACGCTCTTCCGATCT] and Illumina multiplex primer [CAAGCAGAAGACGGCATACGAGATNNNNNNNGTGACTGGAGTTC, where the N stretch corresponds to a six nucleotide index tag]), for 8–16 cycles depending on library strength. Libraries were sequenced using 100 bp single-end (SE; ancient samples) or 100 bp paired-end (PE; modern samples) chemistry on an Illumina HiSeq 2000/2500 platform at The Danish National High-Throughput DNA Sequencing Centre (now Centre for GeoGenetics Sequencing Core).

**Raw read processing and mapping to the reference genome**. Raw reads were first processed using AdapterRemoval2[57] (version 2.3.0); for both SE and PE reads, adapters were removed and N-bases trimmed, and for PE reads, read 1 (R1) and R2 reads were collapsed where possible. Where multiple sequencing lanes of data were generated, trimmed read sets were merged before mapping.

Mapping was performed using BWA-MEM[58] (version 0.7.17-r1188) to an unpublished but significantly improved reference genome of *T. trichiura* (available here: https://github.com/stephenrdoyle/ancient_trichuris/tree/master/02_data). Originally described by Foth et al.[59], the new assembly used here is larger (80.57 Mb vs 75.49 Mb), more contiguous (N50 = 11.3 Mb [vs 0.07 Mb] & N50n = 2 [vs 265]), and in fewer scaffolds overall (*n* = 113 vs 4156) relative to the published version. Three linkage groups are defined via synteny with the *Trichuris muris* genome[59,60] and are designated by the following prefixes in the reference assembly: Trichuris_trichiura_1 (LG1 [X]; X-chromosome), Trichuris_trichiura_2 (LG2; autosome), and Trichuris_trichiura_3 (LG3; autosome). Scaffolds unassigned to linkage groups have the prefix Trichuris_trichiura_00. For the ancient samples, trimmed SE reads were mapped, whereas for the modern samples, trimmed PE and SE reads (merged PE and trimmed SE) were mapped independently and subsequently merged. For all samples, duplicate reads were identified using Picard MarkDuplicates (version 2.5.0; http://broadinstitute.github.io/picard/).

Putative deamination damage (represented by excessive C-to-T and G-to-A substitutions, particularly at the ends of DNA fragments, including sequencing reads) was assessed using PMDtools (version 0.60; https://github.com/pontussk/PMDtools)[61]. This analysis revealed bias in the terminal 2 bp, particularly in the ancient samples; to account for this, the mapped reads were trimmed by 2 bp for all samples (ancient and modern) using bamUtils trimBam[62] (version 1.0.14). Genome coverage was determined using samtools bedcov (version 1.9) in 100 kb windows per scaffold, and sex was determined using the coverage ratio between the sex-linked and autosomal scaffolds. To explore putative causes of mapping variation, we ran Kraken2[63] (version 2.1.2) to estimate the degree of contamination in the raw sequencing reads. This analysis showed that while each sample contained a small degree of contamination evidenced by hits to the Kraken database (minikraken2_v1_8GB), it did not explain overall variation in mapping. Quality control and quantitative comparison between samples were undertaken at each stage of the pipeline and visualised using MultiQC[64] (version 1.8).

**Variant calling**. Variant calling was performed with GATK Haplotype Caller (version 4.1.4.1), first by generating per sample GVCF files (--min-base-quality-score 20 --minimum-mapping-quality 30 ---standard-min-confidence-threshold-for-calling 30), followed by joint genotyping of the sample cohort. Heterozygosity was adjusted from default settings based on estimates from raw reads using GenomeScope[65] (http://qb.cshl.edu/genomescope/). To improve efficiency, each task was split by sample and by scaffold and run in parallel before merging to produce the final call sets.

The raw output was split into mitochondrial and nuclear variants and then split again into single nucleotide variants (SNV) and indels and filtered independently. A

hard filtering approach was employed based on removing relevant tails of variant distributions (typically the upper and/or lower 1%) on the following quality metrics: QUAL, DP, MQ, SOR, FS, QD, MQRankSum, and ReadPosRankSum. Variants were further filtered to ensure they met the following criteria: minimum and maximum alleles = 2; minor allele > 0.02; Hardy Weinberg Equilibrium = >1e-6 (nuclear variants); per sample missingness > 0.8; per genotype depth > 3. Finally, variants were removed if they were found in regions of the genome where the reference genome-guided mapping of kmers (k = 35) was poor or not unique, determined using SNPable (version seqbility-20091110; http://lh3lh3.users.sourceforge.net/snpable.shtml); regions of the genome where overlapping 35-mers mapped uniquely and without 1-mismatch were retained (75.02% of the genome was retained).

The ancient samples underwent an additional round of variant calling using a pool-seq framework, using samtools mpileup and popoolation2 mpileup2sync (--min-qual 20)[66] (version popoolation2_1201) to identify variable sites.

**Population structure analyses**. Broad-scale genetic relatedness between samples and populations was explored by principal component analysis using the R package SNPrelate[67] (version 3.15). These analyses were performed on both mitochondrial and autosomal nuclear variants separately and with and without the colobus and leaf monkey samples, which were outliers in all analyses.

To explore the impact of sample heterogeneity on broad-scale relatedness and population structure, we subsampled the BAM files selecting a single allele at each variant site using ANGSD[68] (version 0.933-102-g7d57642; parameters: -minMapQ 30 -minQ 20 -GL 2 -doMajorMinor 1 -doMaf 1 -SNP_pval 2e-6 -doIBS 1 -doCounts 1 -doCov 1 -makeMatrix 1 -minMaf 0.05), from which pairwise identify-by-state (IBS) matrices were analyses using multidimensional scaling. This approach also used genotype likelihoods per variant site (as opposed to genotypes in the PCA analysis), which are more sensitive to poorer quality and low coverage data.

Genetic differentiation between ancient samples was performed using popoolation2 fst-sliding.pl (--pool-size 1000 --window-size 15000 --step-size 15000 --min-count 1 --min-coverage 2 --max-coverage 2%).

**Phylogenetics of reconstructed and publicly available mitochondrial genomes**. Preprocessed sequence data from modern samples were mapped to published mitochondrial genomes of *T. trichiura* (accession numbers: "KT449826 [https://www.ebi.ac.uk/ena/browser/view/KT449826]", "GU385218 [https://www.ebi.ac.uk/ena/browser/view/GU385218]"and "HG806815 [https://www.ebi.ac.uk/ena/browser/view/HG806815]"[an extract from the *T. trichiura* whole-genome shotgun sequence], *T. suis* ("KT449822 [https://www.ebi.ac.uk/ena/browser/view/KT449822]", "KT449823 [https://www.ebi.ac.uk/ena/browser/view/KT449823]", "GU070737 [https://www.ebi.ac.uk/ena/browser/view/GU070737]"), *T. discolor* ("JQ996231 [https://www.ebi.ac.uk/ena/browser/view/JQ996231]"), *T. ovis* ("JQ996232 [https://www.ebi.ac.uk/ena/browser/view/JQ996232]") and that of the Leaf Monkey, *Trichuris* sp. ("KC461179 [https://www.ebi.ac.uk/ena/browser/view/KC461179]") reporting only the best hit. The published genome with the most hits was then used as a reference in guided assemblies using the mitochondrial baiting and iterative mapping approach, MITObim (version 1.8)[69]. Subsequent manual curation of the assemblies was performed to remove redundant, overlapping sequences at the ends of the mitochondrial contigs, and the starting position was adjusted to the first codon of the COXI gene. For the ancient samples, consensus mitochondrial genome sequences were previously called for ten samples (of 13 in total) that provided 100% bases with coverage to the "KT449826 [https://www.ebi.ac.uk/ena/browser/view/KT449826]" *T. trichiura* reference sequence[13]. Consensus sequences were annotated using the MITOS Webserver[70]. Assembled mitochondrial genomes were aligned together with all *Trichuris* spp. reference genomes available at NCBI using mafft (version 7)[71], with the global-pair setting at 1,000 iterations. A neighbour-joining tree was built using the Jukes-Cantor nucleotide distance model and 1,000 bootstraps in CLC Sequence Viewer 7 (Qiagen). The mitochondrial genome of *Trichinella spiralis* ("NC_002681 [https://www.ncbi.nlm.nih.gov/nuccore/NC_002681]") was used as an outgroup.

**Admixture and population demographic analyses**. Admixture was determined using NGSadmix from the ANGSD package[72]. Briefly, genotype likelihoods were extracted using vcftools (version 0.1.16;--max-missing 1 --BEAGLE-PL), after which NGSadmix (-P 4 -minMaf 0.05 -misTol 0.9) was run over a range of K values from 2 to 10. The optimal value of K was determined by iteratively running NGSadmix as above five times, changing the seed value on each run. The log-likelihoods of each run (all iterations of K = 2–10, s = 1–5) were used to determine the optimal value of K using Evanno's method[73] on the Clumpak web server[74].

Treemix[75] (version 1.12) was performed on the nuclear dataset, which included leaf and colobus monkey samples as outliers. Variants were first filtered using vcftools (--max-missing 1), followed by further pruning to minimise variants in linkage disequilibrium. Customs scripts (ldPruning.sh & vcf2treemix.sh) were modified from https://github.com/speciationgenomics/scripts. The optimal number of migration edges was estimated using the R package OptM (https://cran.r-project.org/web/packages/OptM/).

*f*-statistics were calculated using ADMIXTOOLS[76] qp3Pop (version 435) to calculate outgroup *f*3 data. All combinations of source 1 and source 2 populations

were determined, using either baboon or Honduras samples as the outgroup. Customs scripts (convertVCFtoEigenstrat.sh) were modified from https://github.com/joanam/scripts. Standard error was calculated using a weighted block jackknife, using three blocks. For each comparison, a Z score was determined to test the deviation from 0 (no allele sharing); a Z score of three or greater was deemed significant.

Population demographics were determined using SMC++[77] (version 1.15.2). For each population, variant sites present in all individuals were extracted using vcftools (--max-missing 1), after which smc++ vcf2smc was run per scaffold. Estimated population sizes were fit to the data for all scaffolds using smc++ estimate, which used the nematode *C. elegans* mutation rate of 2.9e-9 mutations per site per generation[78] as a proxy for *T. trichiura* mutation rate, which is currently unknown. Finally, the effective population size per generation was scaled based on an estimated generation time; while unknown precisely, the prepatent period has been estimated to range between 13 to 16 weeks[3] and, therefore, we chose a period of three months or four generations per year. As the ancient samples were a pooled population of eggs rather than individual parasites, they violated the assumptions of the model and were, therefore, removed from the analysis.

The kinship between samples within populations was assessed using vcftools (--relatedness2) and NGSRelate[79].

**Genome-wide genetic diversity analyses.** Genome-wide nucleotide diversity per population and pairwise $F_{ST}$ and Dxy was determined using pixy[80] (version 1.2.6.beta1) in 20 kb non-overlapping sliding windows. The input to pixy is an all-sites VCF containing both variant and non-variant sites, which was derived from the per sample gVCF files generated by GATK (described above). To improve the visualisation of the genome-wide comparisons, analyses were restricted to scaffolds in chromosomal linkage groups, representing 89.83% of the whole genome assembly. Mitochondrial nucleotide diversity of the ancient samples was assessed by generating individual pileup files from the multisample mpileup, followed by analysis using npstat[81] (version 1; -n 1000 -l 15000 -mincov 2 -minqual 20).

The annotation of the unpublished genome assembly used in this analysis was not available at the time of analysis. To identify genes within outlier regions of the genome-wide analyses, we performed a liftover of existing gene models and gene identifiers from the published version of the *T. trichiura* assembly available in WormBase ParaSite[82] (https://parasite.wormbase.org/Trichuris_trichiura_prjeb535/Info/Index/, Version: WBPS15) using liftoff[83] (version 1.5.1). By retaining gene identifiers from the original assembly, cross-validation of gene hits within WormBase ParaSite could be performed. In total, 8,451 of 9,650 gene features (~88%) were transferred; of the genes that did not transfer, ~80% were classified as contamination based on hits to the uniprot_reference_proteome database using DIAMOND[84] (version 0.9.14).

Gene ontology term (GOterms) analysis was performed on the top 5% of $F_{ST}$ outliers using gProfiler[83,85], applying a g:SCS multiple testing correction method and a significance threshold of 0.05, restricting the complete geneset to only genes that had successfully been transferred from the original annotation. GOterms for *T. trichiura* genes can be obtained from WormBase ParaSite using BioMart or can be used directly within gProfiler by selecting the *T. trichiura* genome version and uploading the list of gene identifiers.

Analysis of variation in β-tubulin was performed using vcftools (--site-pi --maf 0.01), using a bed file of exon coordinates derived from manual curation of the gene.

**Reporting summary.** Further information on research design is available in the Nature Research Reporting Summary linked to this article.

## Data availability

Raw sequencing data is available under the European Nucleotide Archive (ENA) study accession "ERP128004". Individual ENA sample accessions are described in Supplementary Data 1. The *T. trichiura* reference genome and assembled mitochondrial genomes are available from https://github.com/stephenrdoyle/ancient_trichuris/tree/master/02_data. The publicly available mitochondrial genomes used for comparative analysis (Accession numbers: "KT449826", "GU385218", "HG806815", "KT449822", "KT449823", "GU070737", "JQ996231", "JQ996232", "KC461179", "NC_017747", "NC_018597", "NC_018596", "NC_028621", "NC_002681") are available from ENA and NCBI. There are no restrictions on data availability.

## Code availability

Custom code to analyse data and reproduce the figures presented is available at https://stephenrdoyle.github.io/ancient_trichuris/ and is archived under a stable DOI at Zenodo[86].

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

## Acknowledgements

We gratefully acknowledge Eske Willerslev, Kurt H. Kjær, and Martin Sikora for helpful discussions during the early stages of the project, Inga Merkyte, Mette Marie Hald, Kirsten Haase, Rikke Simonsen, and Ruben Habraken for access to ancient samples, and Mads Frost Bertelsen for providing samples from baboons. This work was funded by the University of Copenhagen's KU2016 initiative 'The Genomic History of Denmark' (C.M.O.K., M.J.S.), a UKRI Future Leaders Fellowship [MR/T020733/1] (S.R.D.), and the Wellcome Trust [206194] (S.R.D., M.Ber.). X.Q.Z. is supported by the Fund for Shanxi '1331 Project' [20211331-13] and the Special Research Fund of Shanxi Agricultural University for High-level Talents [2021XG001]. For the purpose of Open Access, the author has applied a CC BY public copyright licence to any Author Accepted Manuscript version arising from this submission.

## Author contributions

C.M.O.K., P.N., and M.J.S. designed the study. M.J.S. extracted DNA from ancient samples and prepared NGS libraries for sequencing for all samples. P.N. extracted DNA from modern worm samples. M.Bet., P.J.C., L.P., X.Q.Z., A.S., G.M., G.A.F.S., C.C., B.L., L.-A.T.T., Z.M., S.M.A., and H.N. oversaw the collection of modern worm samples. B.L.F. and M.Ber. provided expertise and advice throughout the study. C.M.O.K., P.N., M.J.S., and S.R.D. analysed and interpreted the results. S.R.D. led and performed the bioinformatics analyses and drafted the manuscript and figures. All authors have read and approved the final version of the manuscript.

## Competing interests

The authors declare no competing interests.
