## [Peer Review File · Nature Communications]

Population genomics of ancient and modern *Trichuris trichiura*REVIEWER COMMENTS

Reviewer #1 (Remarks to the Author):

Doyle et al. present work on the population genomics of the whipworm *Trichuris trichiura*, combining data on ancient and on contemporary specimens of the parasite. The modern samples were collected from human infections (China, Latin America, Sub-Saharan Africa), but also from three species of non-human primates. The authors find that human and baboon worm samples are related, and the highest genetic diversity of parasites localized to Uganda. The authors interpreted their findings in the light of an "Out-of-Africa" hypothesis, whereby human migration has led to the dispersal of *T. trichiura* across the globe.

Comments:

1) in Suppl. Fig. 4 (IBS matrix calculated with ANGSD using all samples), high coverage samples (Uganda, Honduras, China) share more resemblance to each than to low coverage samples from Tanzania and from Cameroon. To me this looks like an artefact of low genome coverage. Please clarify.

2) The ancient samples do not really fit in. If an "Out-of-Africa" hypothesis is correct, the ancient European samples should be closely related to the samples from China. This is not the case, see Suppl. Fig. 4.

3) Relatedness of baboon samples to human samples. Were the baboons born and raised in captivity? How can you be sure that the parasites retrieved from these animals match the parasites of wild baboons?

4) Line 271. Is there any explanation why the population size of *T. trichiura* exemplified by samples from Uganda declined about 10,000 years ago?

5) Line 329. Shrinking population size in ancient European *T. trichiura* population. How can this be explained? The living conditions in European settlements and the absence of medical intervention during the Middle Ages and in the early modern era should have provided favourable conditions for an expansion of *T. trichiura* populations.

Reviewer #2 (Remarks to the Author):

General points

The manuscript presents a large number of whole genome data for *Trichuris trichiura* and represents a large sequencing effort that will be very valuable to the field. However, it is unclear how important the analyses that are presented will be to the field. For example, other studies on modern datasets seem to provide very similar conclusions (e.g. Hawash, M.B.F., Betson, M., Al-Jubury, A. et al. Whipworms in humans and pigs: origins and demography. *Parasites Vectors* 9, 37 (2016). <https://doi.org/10.1186/s13071-016-1325-8>)). It is important to clearly indicate how the new analyses contribute other than simply by providing more sequence data. Also in the paper indicated above samples were presented from China and Uganda (and perhaps the non human primate samples from Denmark), are these the same as those presented in this manuscript? (it is important to identify if any samples have been reported previously in the current manuscript as it might confuse the field)

One of the key issues with the current manuscript is that of whether sampling was sufficient to perform some of the analyses (e.g. were the Chinese samples from one patient?) or to interpret them in the manner presented. With so few samples from each location and relatively few locations (really just three modern locations) the big question would be how representative these samples are even of the countries that are indicated? The title makes a strong statement on using ancient samples but these are all derived from three locations which may well not be representative. It appears that very little has been done with the ancient genomes and it is important that the pipelines used to create these assemblies are carefully considered so as to invalidate some of the comparisons with modern single worms. In some ways the data warrants more than one paper, with one focussed on modern questions and then one with ancient samples

and perhaps being more inclusive of modern multi-parasite samples (e.g. those reported from Cameroon and Tanzania, which are reported then dismissed as being poor quality in this manuscript [they may have been sufficient for some types of analysis such as comparisons of sequence diversity with the ancient samples). It is unclear how variation in the ancient samples was handled when performing some of the analyses in this manuscript).

The discussion does partially address issues with the datasets and explains how many of the analyses are difficult to interpret but perhaps efforts could have been made to obtain more samples including some that the authors have published previously). These could have been driven by preliminary analyses, for example having two or three locations in each of the continents would have given more power to the analyses. Although the effort in sequencing is to be applauded the problems leaves the reader asking, why not change the samples to be sequenced according to the questions being addressed.

There is a statement at the end of the introduction "we provide broad- and fine-scale genetic structure, admixture, and population demographics of geographically and genetically distinct populations. We also explore genome-wide evidence of local adaptation between different human worms and between human and animal worms." . What is meant by fine scale genetic structure? The discussion ought to more explicitly state what was found and how the analysis provides new insights that are different from previous studies.

Specific questions:

1. Very little seems to have been done with samples from Cameroon and Tanzania. The former could have been really useful in terms of the question of forced relocation genetic signatures. It is accepted that these samples were not processed adequately for whole genome analyses but is there enough to ask questions raised in the text of the manuscript.
2. Were the pipelines for modern and ancient samples identical, this is not clear, for example if the ten of the ancient mitochondrial samples were assembled to reference genomes how was a representation of diversity retained in these mixed samples? It would be inappropriate to call a consensus sequence from a sample which would artificially reduce diversity. Equally it is important to be careful about identifying differences between real variation and damage-induced polymorphism. It is appropriate to remove the ends of reads but damage can occur elsewhere in the fragments. How was this controlled for?
3. Modern samples from worms (mostly) but also some faecal specimens which were excluded as insufficient dna was extracted but they could have been used to estimate the degree of variation within an individual versus in a mixed source
4. One of the key difficulties is trying to compare different sample types (mixed versus single worms) where the mixed samples are from different locations to the single worms. Also are there enough locations to be sure of the population genetics, e.g, how variable are the worms across different distances,, are the sites representative of the regions or countries (on what scale)? Please consider the issues related to how representative the data is of the locations and how does sampling or processing differences affect the data.
5. The criteria for genetic analysis of at least 3 times coverage and across 80% of sites in a dataset meant that many samples were discarded including almost all of the ancient samples for nuclear genome analyses. However, on reading the supplemental Table it doesn't appear that these inclusion criteria are met by any of the ancient samples depicted in fig 1d (why are they depicted?).
6. Could the authors comment on the placement of baboon on fig1d compared with the mitochondrial placement on fig1b/c. specifically it appear to be better segregated in the nuclear genome compared with mitochondrial data.
7. For fig 1c, (a small region of fig1b. Are the groups significantly different or just visually segregated?
8. Supp data table 1 the age of sample is given as ancient and modern, should be much more refined for the ancient samples. Similarly the location identifiers appear incomplete and are only offered at a country level (e.g. Cameroon and Ugandan samples), more specific location data is needed (particularly if this data is going to be a foundation for the future). Some of the region annotation appears to be an attribution to an author. Perhaps there should be a latitude-longitude identifier
9. Genome coverage, the text deserves more information, for example the range of coverage not just the mean and this is especially true of the ancient samples where only 2 samples are greater than 2 fold coverage and most are very low coverage
10. The inclusion of the hugely impressive ancient sample DNA within a modern regional analysis

of origin etc seems to hide the potential for more thoughtful analysis of the ancient samples. Yes they are variable and lower coverage than modern but there will almost certainly be a huge amount of interesting signatures in these samples.

11. Some very interesting information on the genetic segregation of samples from the non-human primate (NHP) versus human versus *T. suis* samples is tucked away in supplementary and should perhaps be promoted. The NHP samples in this study were all from zoo environments and need to be treated carefully as it is possible that these worms are non-native to the species that they were sampled from (other NHP specialist perhaps). Again more samples are needed and perhaps some from faeces of co-resident NHP in the zoos.

12. Patterns of admixture> how much is the finding of three ancestral groups a simple consequence of having three modern sites. If there were more sites and samples would the picture be different. An interesting point is made in the discussion about seeking some West African samples for better comparison with Honduras, this ought to be included (were the samples from Cameroon so bad as not to get enough sequence even to test this hypothesis?).

13. How accurate are the estimated population demography changes are with small numbers of locations? What is the confidence level in the results of this analysis and are there any qualifying statements that should be employed in addition to the statement on assumptions? It is a concern that the limited sites in the ancient dataset are used to suggest population contraction where this could be just the nature of the samples (as indicated in the discussion).

14. There is a lot of discussion in the results section around figure 2 which perhaps indicates a lack of confidence in the analyses. Perhaps SF9 needs to be promoted to the main manuscript if it conveys key information. However the legend to this figure is inadequate (as are many other SF legends). I think the proportions of shared variants between samples from the same location are depicted on the left (legend not clear) are the location pairwise estimates all samples from location X compared with variants seen in all samples in location Y? The authors here do comment on the findings "the tenuous due to the low sample numbers and sparse geographic sampling" surely this holds for many of the analyses?

15. Genome wide patterns of selection and adaptation>> SF10 another example where the legend is insufficient e.g. what do the box and whiskers represent. Are the modern samples significantly different to each other, it doesn't look like it. How was mean nucleotide diversity calculated, compared to what?

16. The contracting population comment on ancient DNA associated with Tajimas D might be a result of the handling of the sequence data from a mixed population (this might lead to a loss of rare alleles [processing] reasons) or might reflect the locations.

17. Fig 3 what do the colours mean (not in legend). When considering local adaptation the 50kb sliding window approach might pick up large changes but it will surely hide adaptation in specific genes or parts of genes. The F_{ST} values will determine divergence but there are other measures that could be more powerful to determine selection on open reading frames. Does the inclusion of lots of non-coding sequence will dampen any signal of adaptation. This is a concern especially when the study reports a negative result.

18. Line 339 mentions an analysis of GO terms and refers to sup table 3.4 and 6. Examination of these tables revealed that only a subset of the genes had *C. elegans* orthologues and there were no GO terms listed in these tables. This should be corrected and they should be available for all genes in the Table (even if based upon most likely homologue).

19. Anthelmintic resistance: The analysis again didn't identify anything specific in this study (probably because sampling was opportune not directed). There is something odd about diversity of this locus in the ancient samples (not commented on in the text) (is this due to data processing? Creating a consensus would dampen diversity). Also what about the NHP samples? The baboon and hinduras samples might be a little different than China and Uganda but this would need statistical analysis.

20. Discussion>> The discussion often comments on the limitations of the data sets being analysed, which is appropriate but does call into question whether more data sets need to be analysed to make this study stand out and provide new insights above those already published with selected genetic markers (rather than whole genome data).

21. Line 470 the comment that this data establishes a framework for genetic analysis should be toned down and indicate that it is a starting point.

22. Data availability: The comment in materials and methods for the calling of 10 of the ancient sample mitochondria stimulate me to examine the PlosOne paper. Unfortunately in this there was no link to the accession files for this data. It is good to see that the raw read data is going to be

available (will this be for all sequence data including those that didn't make it into the analysis?). It is important that the assemblies created as part of the current study also be made available.

Reviewer #3 (Remarks to the Author):

This paper reports the first population genomic analysis of the important human parasite *Trichuris trichiura*, including samples that span the entire globe and the entire past millennium. Despite various technical challenges, including irrevocably pooled samples and low coverage for some samples, this is one of the most comprehensive population genomic surveys of a human helminth to date. These results will facilitate future efforts to understand and control this parasite, including monitoring and predicting how worm populations respond to drugs.

The samples vary widely with respect to collection methodology and quality of the data. The details of these differences are largely explained, although it's not always clear which samples originate from the same patient, a crucial detail since worms from the same host are more likely to be closely related and may not reflect the full diversity of their population. The China samples all come from the same patient, and at least some of the Uganda samples come from the same patient, but not all? It would be good to note all of this explicitly. Samples can be classified as pooled, single worm low coverage, and single worm high coverage, but it's not obvious from the main text which worms are which; can this be indicated more clearly?

Because of the differences in sample quality, there are caveats to analyzing all samples jointly. For example, in PCA, differences in apparent heterozygosity owing to differences in coverage could drive the results. It would be good to generate a PCA using haploidized data (drawing one allele at random from all sites in each sample) to test for this effect. Similarly, using pooled data probably violates the assumptions of NGSadmix, though the results appear to be robust to this since the ancient samples still form a distinct group along with the baboon samples from the same continent.

Sex chromosomes and autosomes have different demographic histories and ideally should be analyzed separately. It's not explicitly stated which scaffolds in the figures (e.g. Fig 3) are sex-linked, and it would be good to clarify this. Also, genome-wide analyses like PCA and summary statistics like π and Tajima's D should be done separately for autosomes and sex chromosomes. This can reveal sex differences in mating success or migration, and also autosomes may be more subject to phenomena like introgression from other species. An obvious thing to test is whether π on X-linked scaffolds is 75% of π on autosomes, which should be the null hypothesis. Why are there no genome-wide depictions of π , as there are for F_{st} (Fig 3) and Tajima's D (Supp Fig 11)?

A PCA of mitochondrial data (Fig 1 b & c) is unusual and violates the assumptions of PCA since all mtDNA variants are in linkage disequilibrium with each other and are not independent. The mitochondrial phylogeny (Supp Fig 6b) is much clearer and evolutionarily valid. This, or a simplified version of it, should be in the main text instead of the mtDNA PCA.

Tajima's D is not biologically meaningful with fewer than four sequences, since all allele frequencies are identical. The authors note the high levels of positive Tajima's D on the two X chromosomes from baboon-derived worms; in fact Tajima's D is effectively impossible to calculate in this scenario and shouldn't even be mentioned or included in Supp Fig 11b.

It would be nice to see mean heterozygosity estimates for each sample, perhaps in a supplemental table or figure. The decrease in π from Uganda to China to Honduras is one of the more salient results, and it would be good to know how robust it is. Do all Uganda samples, considered individually, have higher heterozygosity than all Honduras samples, for example? Heterozygosity may be confounded with coverage, so maybe plot heterozygosity versus coverage in order to show any patterns.

There is a lot of literature comparing the relative utility of F_{st} and D_{xy} . Since opinions differ on these statistics, and it's clear that they can reveal different evolutionary phenomena, can D_{xy} also be shown?

In Figure 1, is it possible to use the same symbols and colors in part A and in the later parts? For both the map and the PCA, it would be nice to see at a glance which samples are from which country and which ones are from human or other host species, etc., and also which are pooled.

In Fig 2b, are there 2 or 3 migration edges? There are three colored lines but the caption says two.

Supp Figure 12a: Do the blue bars represent individual SNPs? If so, " π " is an unusual way to report them, since π usually refers to segments more than 1bp long. Can this be stated as expected heterozygosity or minor allele frequency?

REVIEWER COMMENTS

Reviewer #1 (Remarks to the Author):

Doyle et al. present work on the population genomics of the whipworm *Trichuris trichiura*, combining data on ancient and on contemporary specimens of the parasite. The modern samples were collected from human infections (China, Latin America, Sub-Saharan Africa), but also from three species of non-human primates. The authors find that human and baboon worm samples are related, and the highest genetic diversity of parasites localized to Uganda. The authors interpreted their findings in the light of an “Out-of-Africa” hypothesis, whereby human migration has led to the dispersal of *T. trichiura* across the globe.

Comments:

R1.1: in Suppl. Fig. 4 (IBS matrix calculated with ANGSD using all samples), high coverage samples (Uganda, Honduras, China) share more resemblance to each than to low coverage samples from Tanzania and from Cameroon. To me this looks like an artefact of low genome coverage. Please clarify.

RESPONSE: We initially used the IBS matrix to show some clustering between the very low coverage samples (which ANGSD is suitable for) from Tanzania and Cameroon. Specifically, we used parameters in ANGSD to choose a single read at random to compare variation which, in part, accounts for differences in coverage as it normalises all samples to a coverage of 1x. However, reevaluation of these data after consideration of this comment does suggest there is a missingness bias, in which the low coverage samples have a higher missingness and, therefore, appear distinct.

To address this in the manuscript, we have decided to remove these data.

R1.2: The ancient samples do not really fit in. If an “Out-of-Africa” hypothesis is correct, the ancient European samples should be closely related to the samples from China. This is not the case, see Suppl. Fig. 4.

RESPONSE: Given that we believe the Suppl. Fig. 4 is perhaps biased by missingness as per the previous comment, we have removed these data from the manuscript. We describe the relationship between the ancient samples and others, in the context of relative nucleotide diversity and levels of allele sharing, elsewhere in the manuscript.

R1.3: Relatedness of baboon samples to human samples. Were the baboons born and raised in captivity? How can you be sure that the parasites retrieved from these animals match the parasites of wild baboons?

RESPONSE: At the Copenhagen Zoo where these samples were collected, a baboon colony has been maintained for decades and of which the population size of baboons is controlled via euthanasia each year. There are no endemic *Trichuris* infecting humans in Denmark and so while it is likely that this captive population maintains baboon-infective *T. trichiura*, the parasite population would have been established from parasites infecting wild-caught baboons.

We do not claim that the baboon-infective parasites we have sampled reflect all parasites that infect baboons in the wild. This statement is true for the human parasites sampled, which are a proxy for the populations from which they are derived, but certainly do not describe the complete diversity present in those populations. If we sampled baboon-infective parasites from the wild, we would likely find new genetic diversity and this genetic diversity may show genetic structure based on where they were sampled, i.e., parasites from East Africa would likely be genetically distinguishable from West Africa. However, the key finding is that the diversity of baboon-infective parasites is not significantly different from the human-infective parasite populations, even when they are broadly sampled throughout the world. This is in stark contrast to the other non-human primates we sampled - these were highly genetically distinct from both human and baboon-infective parasites, as we show in Supplementary Figures 5 a and b.

R1.4: Line 271. Is there any explanation why the population size of *T. trichiura* exemplified by samples from Uganda declined about 10,000 years ago?

RESPONSE: This is an important observation and something we thought about for some time. Prompted by this comment, we found literature that identified extreme changes in weather conditions approximately 15-17,000 years before the present that affected the Afro-Asian monsoon region, which includes Uganda. Described as “one of the most extreme and widespread megadroughts of the past 50,000 years” (Stager 2011 Science; PMID: 21350124), one consequence was in desiccation of Lake Victoria (Johnson et al 1996 Science; PMID: 8688092), the world’s largest lake and one half of which is in Uganda.

Trichuris trichiura is broadly distributed in tropical regions where warm, moist conditions favour its development in the soil prior to reinfection. Rapid, severe and sustained drought would have likely put huge pressure on this environmentally vulnerable stage in the life-cycle, and it is not difficult to imagine a major bottleneck in the population as a consequence.

R1.5: Line 329. Shrinking population size in ancient European *T. trichiura* population. How can this be explained? The living conditions in European settlements and the absence of medical intervention during the Middle Ages and in the early modern era should have provided favourable conditions for an expansion of *T. trichiura* populations.

RESPONSE: We agree with the reviewer that living conditions and the absence of medical interventions would have provided favourable conditions for *T. trichiura* infections in Europe. We have sampled parasites from a range of time points over the last 1000 years, and there are coprological findings of much older parasite infections available. What is unclear is the relative population size of these parasites, especially when compared to modern populations in endemic countries today.

We provide one hypothesis for why we observe a difference in parasite populations, and this is firmly supported by the developmental biology of the parasite. As we describe in the discussion of the manuscript, parasite development is temperature-dependent, and below 15C degrees, development arrests. Therefore in higher latitudes such as in Northern Europe, there is a maximum of one generation per year, which is in stark contrast to 3-4 generations per year in warmer climates. Thus, while it is impossible to provide a precise estimate of the population size of the Northern European populations with the data we have collected, it is fair to assume that there would be differences in the population size based on environmental conditions that favour (or not) development and therefore transmission.

Reviewer #2 (Remarks to the Author):

General points

The manuscript presents a large number of whole-genome data for *Trichuris trichiura* and represents a large sequencing effort that will be very valuable to the field.

R2.1: However, it is unclear how important the analyses that are presented will be to the field. For example, other studies on modern datasets seem to provide very similar conclusions (e.g. Hawash, M.B.F., Betson, M., Al-Jubury, A. et al. Whipworms in humans and pigs: origins and demography. *Parasites Vectors* 9, 37 (2016).

<https://doi.org/10.1186/s13071-016-1325-8> [doi.org])). It is important to clearly indicate how the new analyses contribute other than simply by providing more sequence data.

RESPONSE: Despite some overlapping aims, we strongly argue that our manuscript is far more comprehensive than “simply by providing more sequence data”. We make some important distinctions from Hawash and colleagues, including:

(i) Hawash focused only on two partial mitochondrial genes (*rrnL* and *nad1*), whereas here, we present genomic analyses based on whole mitochondrial genomes and whole nuclear genomes. As we describe in the comment further down, we explicitly demonstrate the resolving power of comparing variation from the nuclear genome against variation from whole mitochondrial genomes, let alone only partial mitochondrial fragments in Hawash et al. ;

(ii) we address questions using nuclear data such as fine-scale genetic relatedness, admixture and kinship that simply cannot be addressed using fragmented mitochondrial data presented in Hawash et al. Further, our publicly available data has the potential to be added to and mined further in new ways, whereas the utility of further analysis of the partial mitochondrial genes from Hawash et al. is very limited; and

(iii) we provide analyses of ancient DNA, compared against modern DNA samples, for the first time. To reiterate, these are the oldest whole-genome sequencing data generated for a eukaryotic pathogen to date.

R2.2: Also in the paper indicated above samples were presented from China and Uganda (and perhaps the non human primate samples from Denmark), are these the same as those presented in this manuscript? (it is important to identify if any samples have been reported previously in the current manuscript as it might confuse the field)

RESPONSE: In total, only five samples were the same between studies, chosen on purpose as they were available (see other comments on the challenge of obtaining samples in the first place) and to provide greater resolution on these existing samples. The continuity of these samples between this manuscript and previous reports is the use of the same sample ID in the Supplementary Table 1. All other samples were unique to each study.

Nonetheless, no samples (neither here or elsewhere) have been analysed using whole-genome sequencing to date, apart from a single sample from Ecuador used in the original description of genome assembly (Foth et al 2014 Nat Gen; PMID: 24929830), which has not been used here.

R2.3: One of the key issues with the current manuscript is that of whether sampling was sufficient to perform some of the analyses (e.g. were the Chinese samples from one patient?) or to interpret them in the manner presented. With so few samples from each location and relatively few locations (really just three modern locations) the big question would be how representative these samples are even of the countries that are indicated?

The title makes a strong statement on using ancient samples but these are all derived from three locations which may well not be representative. It appears that very little has been done with the ancient genomes and it is important that the pipelines used to create these assemblies are carefully considered so as to invalidate some of the comparisons with modern single worms. In some ways the data warrants more than one paper, with one focussed on modern questions and then one with ancient samples and perhaps being more inclusive of modern multi-parasite samples (e.g. those reported from Cameroon and Tanzania, which are reported then dismissed as being poor quality in this manuscript [they may have been sufficient for some types of analysis such as comparisons of sequence diversity with the ancient samples]). It is unclear how variation in the ancient samples was handled when performing some of the analyses in this manuscript).

RESPONSE: The reviewer asks a very valid question - is the sampling sufficient? The simple answer is that there could always be more samples, from more appropriate locations, to address additional questions. The primary aim of this manuscript was to provide a broad-scale view of the genetic diversity of this human parasite, which is an essential first step towards understanding the variation that is present and importantly, how this variation may be used to inform strategies relevant to the control of the parasite, currently estimated to infect 500 million people worldwide. We argue we have achieved this primary aim and, together with a sampling of non-human primates and a unique opportunity to sample ancient parasites (by far, the oldest eukaryotic pathogens sequenced to date), we extend these aims to address other relevant life-history traits of the parasite.

Despite infecting 500 million people worldwide, these parasites are in fact quite difficult to collect. Some surveillance campaigns monitor for the presence of parasites in faeces, but these are not routinely archived. The adult parasites are partially intracellular and collection is more difficult and only performed by “expulsion” after specific drug treatments, which again are relatively rare. In addition, worms are expelled over a period of two weeks which makes collection even more challenging. Thus, parasites available for academic use are not as common as one might expect. It is clear from the samples we present that these were opportunistically obtained from material already available, made possible via an assemblage of a diverse group of collaborators many of whom work in endemic countries and therefore had access to samples. It is simply not possible to perform these studies of “global diversity” any other way, particularly on a parasite classed as “neglected” despite such a large burden of infection throughout the world.

The reviewer also sensibly asks, how representative [are] these samples? Again, the answer is clearly we don't know how representative they are of a country or region if one is to consider the diversity of only a small handful of parasites among an uncountable number of parasites from that region, further confounded by the fact that helminth parasites are generally extremely genetically diverse. However, this is not the question we have asked or

tried to address. They are, however, a good proxy for country/regional representation when we do not have any other comparable data and for making broad-scale inferences in line with our primary aims.

We agree with the reviewer that it was not clear which parasites came from the same or different hosts. We intentionally removed host information from the study due to ethical reasons, however, we have partially deidentified this information and now provide it in Supplementary Table 1 under the column "SOURCE_DETAIL". Because the reviewer asked, we can confirm that the Chinese samples came from five different patients.

Ancient samples were analysed using approaches that were appropriate for the sample type. They were not used when, for example, the analysis required data from individuals. Hence, they did not feature in some aspects of the analyses. To remind the reviewer (actually addressed in detail below when asked specifically about this), the mitochondrial genomes of the ancient samples were assembled and used only in the phylogenetic analyses presented in Supplementary Figure 5b; any other analyses using the ancient samples made use of genetic variation in the same way as the modern samples.

Regarding sample handling, we have provided all of the code used to analyse all of the data, including the ancient samples. While this is technically expected of a genomics manuscript by the journal, this is rarely delivered in the extent and detail that we provide. We have done so to both provide transparency of what we have done, and to enable others to make use of and build upon our analysis. This forms part of our commitment to provide a framework upon which future genomic epidemiological studies of soil-transmitted helminths can be based.

R2.4: The discussion does partially address issues with the datasets and explains how many of the analyses are difficult to interpret but perhaps efforts could have been made to obtain more samples including some that the authors have published previously). These could have been driven by preliminary analyses, for example having two or three locations in each of the continents would have given more power to the analyses. Although the effort in sequencing is to be applauded the problems leaves the reader asking, why not change the samples to be sequenced according to the questions being addressed.

RESPONSE: As we describe in detail above, while it would have been ideal to obtain more samples from different populations, collecting any samples at all is not easy. We strongly believe that our manuscript will serve as a platform to enable new studies to undertake targeted sampling to address some of the outstanding questions we raise and to fill in the geographical gaps to provide a more fine-grained understanding of genetic diversity within

countries or regions. The fact that we have provided all of our data and code without restriction is a significant step forward towards building on this work.

R2.5: There is a statement at the end of the introduction “we provide broad- and fine-scale genetic structure, admixture, and population demographics of geographically and genetically distinct populations. We also explore genome-wide evidence of local adaptation between different human worms and between human and animal worms.” . What is meant by fine scale genetic structure? The discussion ought to more explicitly state what was found and how the analysis provides new insights that are different from previous studies.

RESPONSE: It is clear from our analyses that some populations are very clearly genetically distinct, typically at the continent scale. We consider this to be broad-scale genetic structure. However, we also identify much more subtle differences in genetic structure between more closely related populations, and use a number of different approaches to tease this apart. This would be considered fine-scale genetic structure.

Each paragraph of our discussion is focused on a major finding, and places these findings in the context of previous research and/or parasite biology. These include: (i) the fact that we could sequence DNA from eggs up to 1000 years old, (ii) population decline in Northern Europe, (iii) global dispersal, (iv) differences between human and NHP samples, and zoonotic reservoirs, (v) anthelmintic resistance.

Specific questions:

R2.6: Very little seems to have been done with samples from Cameroon and Tanzania. The former could have been really useful in terms of the question of forced relocation genetic signatures. It is accepted that these samples were not processed adequately for whole genome analyses but is there enough to ask questions raised in the text of the manuscript.

RESPONSE: The reviewer is correct, very little was done with the samples from Cameroon and Tanzania. Unfortunately, the genome coverage was so low for both of these populations it was very difficult to do much at all with them. This is indicated in Supplementary Table 2, which shows the low genome coverage and high variant missingness per individual. We included them to reflect that we tried and failed, but that we likely know the reason why. We felt that this was useful information to include for future studies.

R2.7: Were the pipelines for modern and ancient samples identical, this is not clear, for example if the ten of the ancient mitochondrial samples were assembled to reference genomes how was a representation of diversity retained in these mixed samples? It would be inappropriate to call a consensus sequence from a sample which would artificially reduce diversity.

RESPONSE: There were subtle differences between the pipelines used to process the ancient and modern samples. Some of these differences related to the fact that the library preparation and sequencing were different - ancient samples were processed using established protocols for ancient DNA, including sequencing with single-end reads due to fragmented state of ancient DNA degrading over time, whereas modern samples were processed using standard library preparation methods and paired-end sequencing. This difference required different mapping pipelines to be used, for example. Other differences in the analysis of the modern and ancient samples were based on whether those samples were single worms or pooled worms (eggs). All ancient samples were pooled eggs, whereas almost all modern samples were single worms. Some analyses were inappropriate to run on pooled samples, and so they were run only on the single worms.

Regarding the assembly of the mitochondrial genomes for the phylogenetic analyses, we completely agree that a proportion of the variation within the pooled samples is lost by making a consensus sequence. However, this is a very common process applied to a mixed sample in which a consensus sequence is used/needed. Whether that is the original human genome reference assembly (derived from DNA of multiple people), the collapsing of a diploid genome during genome assembly into a haploid reference, or analysis of a mixed worm sample used here, it is very common that the most frequent or consensus nucleotide is used. Here, this was performed for the phylogenetic analysis, as the tools used to analyse the data and generate the phylogeny required a single sequence per sample. Please note that these consensus sequences were ONLY used in the phylogenetic analysis presented in Supplementary Figure 5 b; all other analyses of ancient sample diversity DID retain the diversity within the data as variant frequencies were used.

R2.8: Equally it is important to be careful about identifying differences between real variation and damage-induced polymorphism. It is appropriate to remove the ends of reads but damage can occur elsewhere in the fragments. How was this controlled for?

RESPONSE: We completely agree that it is important to distinguish between real variation and damage-induced polymorphism, so we specifically looked for signatures of this damage in all samples in our dataset. We followed established pipelines for the detection of damage commonly found in ancient DNA. These signatures of damage are commonly found in the ends of the reads and are age-dependent, whereby there is a positive relationship between

damage and sample age. Compared to many ancient DNA studies focused on the analysis of DNA 10s-of-thousands of years old or older, our ancient samples up to 1000 years old are relatively “young” and, therefore, the signatures of DNA damage we found were minor. We examined the profiles of damage throughout the reads (examples shown in Supplementary Figure 2), and indeed found evidence of damage within the first two base pairs; this damage, albeit minor and only found in a relatively low percentage of reads, provided the rationale for removing these bases from all reads.

Regarding damage to the reads beyond the first two base pairs, there were no obvious differences in the damage profiles between the ancient and modern samples. Considering the very low frequency of damage overall (even in the 5' of reads that were removed), analyses such as phylogenetic placement of ancient mitochondrial genomes would almost certainly be unaffected, given a consensus sequence was used. Given the lack of obvious impact, we made no attempt to control for this further.

R2.9: Modern samples from worms (mostly) but also some faecal specimens which were excluded as insufficient dna was extracted but they could have been used to estimate the degree of variation within an individual versus in a mixed source

RESPONSE: The reviewer is correct that most of the modern samples were single adult worms (n = 34), whereas 10 samples were from pools of eggs collected from faeces. While we agree that this is a very sensible idea - to measure diversity on an individual vs pooled - the samples or sampling design was not appropriate for that question. Ideally, both adults and eggs should be collected from the same host to begin to address such a question, however, we did not have any of these paired sample sets. Our data is further limited to address such a question as we do not have pooled AND single worm samples from the same population and, therefore, would not be able to disentangle “pooled vs single worm” from “population A vs population B” diversity.

R2.10: One of the key difficulties is trying to compare different sample types (mixed versus single worms) where the mixed samples are from different locations to the single worms. Also are there enough locations to be sure of the population genetics, e.g, how variable are the worms across different distances,, are the sites representative of the regions or countries (on what scale)? Please consider the issues related to how representative the data is of the locations and how does sampling or processing differences affect the data.

RESPONSE: The reviewer is correct - our sampling is confounded in one sense in that the sample type (pooled vs single worm) is a fixed variable that sometimes differs per geographical region. However, we would like to remind the reviewer that the only pooled

samples were from Tanzania and Cameroon, and these were largely excluded from analyses due to low sequence coverage, and the ancient samples. The remaining samples were single worms. As discussed above and in the manuscript, we argue that the sampling is satisfactory to establish the broad-scale genetic differences of the parasite as was our primary aim.

R2.11: The criteria for genetic analysis of at least 3 times coverage and across 80% of sites in a dataset meant that many samples were discarded including almost all of the ancient samples for nuclear genome analyses. However, on reading the supplemental Table it doesn't appear that these inclusion criteria are met by any of the ancient samples depicted in fig 1d (why are they depicted?).

RESPONSE: The reviewer is correct, in that most (15/17) ancient samples are excluded due to low coverage and, perhaps counterintuitively, the two ancient samples that were included also showed low coverage with both having a mean nuclear coverage of less than 3. However, Supplementary Table 2 presents the mean coverage throughout the genome, and our criteria for inclusion of genetic data was 3x coverage and 80% samples included "per variant site". As coverage is not evenly distributed throughout the genome, many individual sites for the ancient samples still do in fact pass the threshold for inclusion. This is evident in the number of variants retained in each analysis, dependent on the number and type of samples used; although we identified 6,933,531 nuclear variants in total, only 2,631,365 variants were used in nuclear PCA (Figure 1d), a significant reduction largely driven by the inclusion of the low-coverage ancient samples.

R2.12: Could the authors comment on the placement of baboon on fig1d compared with the mitochondrial placement on fig1b/c. specifically it appear to be better segregated in the nuclear genome compared with mitochondrial data.

RESPONSE: The reviewer is correct, in that the placement of baboons is much better discriminated using the nuclear dataset (Fig 1d) than the mitochondrial dataset (Fig 1b,c). This is simply because there is much greater resolving power using the nuclear data containing 2,631,365 variants, compared to the mitochondrial data containing 802 variants. This nicely demonstrates that whole-genome sequencing provides significantly greater resolution than single genes or even whole mitochondrial genomes. This level of resolution will likely be needed as more data is generated, for example, within a country or region.

R2.13: For fig 1c, (a small region of fig1b. Are the groups significantly different or just visually segregated?

RESPONSE: In Figures 1b, c and d, we have used principal component analysis (PCA) to differentiate samples based on their genetic diversity. PCA is a common dimensionality reduction technique to transform complex multivariate data into components that are ranked based on the proportion of variation each component can explain in the data. Irrespective of the data input, PCA is a useful way to explore the data and to understand the degree to which the structure in the data can be explained. Here we use it for visualisation and have not applied any statistical testing to test whether the groups are different. To our knowledge, statistical testing is not applied in PCA. Given the groups are not overlapping, they would most certainly be significantly different to one another, and is supported by the fact that PC1 and PC2 explain as much as 57% of the variance in the data.

R2.14: Supp data table 1 the age of sample is given as ancient and modern, should be much more refined for the ancient samples. Similarly the location identifiers appear incomplete and are only offered at a country level (e.g. Cameroon and Ugandan samples), more specific location data is needed (particularly if this data is going to be a foundation for the future). Some of the region annotation appears to be an attribution to an author. Perhaps there should be a latitude-longitude identifier

RESPONSE: We do use “ancient” and “modern” to differentiate between the two groups (“Age_ID”), however, we agree that more information would be useful.

In response to this comment, we have updated Supplementary Table 1 to be more explicit about ancient sample age, and filled in regional information about the sampling locations.

R2.15: Genome coverage, the text deserves more information, for example the range of coverage not just the mean and this is especially true of the ancient samples where only 2 samples are greater than 2 fold coverage and most are very low coverage

RESPONSE: We agree with the reviewer that an understanding of genome coverage is important. This is why we have provided the mean coverage per sample for all in Supplementary Table 2, reported the average coverages in the text, and were explicit about coverage cutoffs in the text and code used. However, we feel that adding the range estimates (already provided in Supplementary Table 2) to the main text does not add anything insightful, but does make the sentence much longer and more complicated. For example, the original:

“We have generated whole-genome sequencing data from 44 modern and 17 ancient samples (**Supplementary Table 1**), resulting in an average coverage of 9.31× and 0.66× of

the nuclear genomes and 613× and 95× of the mitochondrial genomes, respectively (**Supplementary Table 2**).”

And with ranges:

“We have generated whole-genome sequencing data from 44 modern and 17 ancient samples (**Supplementary Table 1**), resulting in an average coverage of 9.31× [range = 0.003X - 31.07X] and 0.66× [0.006X-2.95X] of the nuclear genomes and 613× [0.328x - 4553.04x] and 95× [1.02x-192.76x] of the mitochondrial genomes, respectively (**Supplementary Table 2**).”

We are happy for the editor to suggest if this level of detail is needed in the main text.

R2.17: The inclusion of the hugely impressive ancient sample DNA within a modern regional analysis of origin etc seems to hide the potential for more thoughtful analysis of the ancient samples. Yes they are variable and lower coverage than modern but there will almost certainly be a huge amount of interesting signatures in these samples.

RESPONSE: This comment prompted us to have a deeper look at the ancient genetic diversity. In doing so, we now describe patterns of shared genetic diversity between ancient populations, based on mitochondrial DNA diversity. There is some degree of isolation by distance; populations that are physically close are clearly more genetically related, especially those in very close proximity (e.g. COG/COA or KAM/ZWO), whereas there is very little genetic sharing between Denmark/Holland and Lithuania (VIL). Within Denmark, VIB (Viborg) is an outlier despite the close proximity to other populations from Denmark, however, it is a much older sample with close to 500 years between these samples and the next sampled population. Broadly, these data demonstrate parasites are likely shared across space, and to some degree, across time.

Given parasites were in effect, sampled across time, we sought to explore how genetic diversity changes over time. In the first version of the manuscript, we presented data based on two ancient genomes that suggested that these populations were undergoing population contraction relative to the modern populations - this has been commented on a number of times in this document. We argued that there is a sensible explanation for this; that the environmental conditions do not favour persistence in northern Europe. Analysis of nucleotide diversity data across time with all of the ancient samples provides a consistent picture supporting our initial observation; that genetic diversity is decreasing indicative of a declining population in northern Europe, well before modern interventions. These data are independent of the first analyses, and thus, provides further support for our hypothesis.

We present these findings and a new Figure 2 in the main text of the manuscript.

R2.18: Some very interesting information on the genetic segregation of samples from the non-human primate (NHP) versus human versus *T. suis* samples is tucked away in supplementary and should perhaps be promoted. The NHP samples in this study were all from zoo environments and need to be treated carefully as it is possible that these worms are non-native to the species that they were sampled from (other NHP specialist perhaps). Again more samples are needed and perhaps some from faeces of co-resident NHP in the zoos.

RESPONSE: We can't comment on the ability of these specific parasite isolates obtained from NHP and their ability to infect other NHP including humans. We can, however, be confident in the degree of genetic differentiation within and between human-infective parasites and the NHP-infective parasites samples; here, human and baboon samples were genetically indistinguishable relative to the colobus and leaf monkey parasite samples, which were not only distinct relative to humans/baboons but also to each other. Although it is interesting to speculate how permissive *Trichuris* spp are in their ability to infect multiple different NHP, our focus is on potential zoonoses and reservoirs of infection that will prevent the control of human-infective parasites. The key experiment to address this is to sample parasites specifically from humans and NHPs that co-inhabit; however, this would require an entirely new sampling framework involving new study collaborators, new regulatory requirements and sequencing, all of which combined could easily exceed 12 months of work or more. Such an experiment is, therefore, beyond the scope of this manuscript.

R2.19: Patterns of admixture> how much is the finding of three ancestral groups a simple consequence of having three modern sites. If there were more sites and samples would the picture be different. An interesting point is made in the discussion about seeking some West African samples for better comparison with Honduras, this ought to be included (were the samples from Cameroon so bad as not to get enough sequence even to test this hypothesis?).

RESPONSE: Considering the clear genetic structure presented in figure 1, it is not necessarily surprising that the admixture analyses also supported three groups that coincided with the three populations. However, what is revealed is more fine-grained detail, for example, it reinforced diversity in Uganda and that European and China diversity is a subset of the African diversity. If there were more sites, we would find both more genetic structure and

more patterns of admixture between populations. This is not a controversial idea and does not invalidate the results presented.

We did in fact seek out new West African samples from our collaborative network, however, we were unable to obtain any single worm samples that would have been necessary to understand the admixture between West Africa and South/Central America. The samples from Cameroon were sufficiently bad that this could not be tested. It is unfortunate but does emphasise how difficult these samples are to obtain, and how valuable the presented sequencing data is.

R2.20: How accurate are the estimated population demography changes are with small numbers of locations? What is the confidence level in the results of this analysis and are there any qualifying statements that should be employed in addition to the statement on assumptions? It is a concern that the limited sites in the ancient dataset are used to suggest population contraction where this could be just the nature of the samples (as indicated in the discussion).

RESPONSE: The population demographic analyses provided are performed on data from each site independently. These analyses can be performed using as few as a single sample, eg PSMC (PMID: PMC3154645). We have used a modern analytical framework - SMC++ - that can use multiple samples per population to estimate demography.

The ancient samples are not included in the population demography analyses presented in figure 2, precisely because they violate the requirements of the SMC++. We do hypothesise that the ancient samples are undergoing population contraction to some degree, at least in contrast to the modern samples. This is based not only on a few data points but consistent genome-wide signatures to this effect, as well as new data showing a decline in genetic diversity over time in the ancient samples. As we discuss, this hypothesis is consistent with the underlying biology of the organism and its developmental rate at different temperatures. Of course, having more samples collected to test this hypothesis from a range of environmental conditions would be ideal, but unfortunately is beyond the scope of what is possible here.

R2.21: There is a lot of discussion in the results section around figure 2 which perhaps indicates a lack of confidence in the analyses. Perhaps SF9 needs to be promoted to the main manuscript if it conveys key information. However the legend to this figure is inadequate (as are many other SF legends). I think the proportions of shared variants between samples from the same location are depicted on the left (legend not clear) are the location pairwise estimates all samples from location X compared with variants seen in all

samples in location Y? The authors here do comment on the findings “the tenuous due to the low sample numbers and sparse geographic sampling” surely this holds for many of the analyses?

RESPONSE: The reviewer makes a note in the following comment that “a lot of discussion... indicates a lack of confidence in the analyses”, however, we argue that we have been very forthcoming, beyond what is typically reported, on our degree of confidence and qualifying statements. We explicitly acknowledge that we cannot be precise in the generation time and mutation rate, as these both cannot be determined easily and may vary between regions of the world, due to additional variables such as climate that would influence development (as we also discuss in the discussion in regard to the ancient samples).

To address this comment, we have made efforts to more comprehensively describe the supplementary figure in the legends (and made updates to other supplementary figure legends where appropriate).

R2.22: Genome wide patterns of selection and adaptation>> SF10 another example where the legend is insufficient e.g. what do the box and whiskers represent. Are the modern samples significantly different to each other, it doesn't look like it. How was mean nucleotide diversity calculated, compared to what?

RESPONSE: We agree that this could have been more descriptive.

In response to this and other comments regarding nucleotide diversity, we have decided to present the figure completely differently. We now show nucleotide diversity genome wide, and then summarise the distribution of the data using a density plot together with the mean. We feel this provides greater context for the variation in diversity throughout the genome, but also, enables comparison between different sample groups.

R2.23: The contracting population comment on ancient DNA associated with Tajimas D might be a result of the handling of the sequence data from a mixed population (this might lead to a loss of rare alleles for technical [processing] reasons) or might reflect the locations.

RESPONSE: Based on this comment and that of another reviewer, we have decided to remove the Tajima's D analyses from the manuscript. It is possible that there are some technical differences that we cannot control. In response to another of this reviewer's comments to explore the ancient DNA samples, we find an independent genetic signal supporting the population contraction of ancient samples.

R2.24: Fig 3 what do the colours mean (not in legend). When considering local adaptation the 50kb sliding window approach might pick up large changes but it will surely hide adaptation in specific genes or parts of genes. The F_{ST} values will determine divergence but there are other measures that could be more powerful to determine selection on open reading frames. Does the inclusion of lots of non-coding sequence will dampen any signal of adaptation. This is a concern especially when the study reports a negative result.

RESPONSE: The reviewer is correct, the colours were not described; we have updated the legend to indicate the colours represent different scaffolds in the genome assembly. To simplify this further, we have changed the colour scheme of the figure so that scaffolds have alternating colours, rather than a colour spectrum as originally presented. We also now indicate the three linkage groups and the sex-linked scaffolds.

The reviewer is also correct in that 50 kb windows are less sensitive to fine-scale changes such as those in individual genes. We have reduced this to 20 kb, which provides some additional resolution; we did reduce the window size further, however, it simply increased the noise. We also agree that there are more sensitive approaches to detecting selection, particularly when a high-quality annotation is available. However, this is not the case here. The version of the assembly used was unannotated in the early stages of analysis, as it was (and in fact still is) being improved. To provide some annotation to work with, we transferred the annotations from the previous more fragmented genome assembly (Foth et al 2014) to the new assembly. While this provides a means to ask “what genes are present?” with some confidence, is it insufficient to be used to detect nucleotide-level resolution changes to say specific things about individual genes. Hence, we have focused on broader genome-wide visualisations and analyses.

R2.25: Line 339 mentions an analysis of GO terms and refers to sup table 3.4 and 6. Examination of these tables revealed that only a subset of the genes had *C. elegans* orthologues and there were no GO terms listed in these tables. This should be corrected and they should be available for all genes in the Table (even if based upon most likely homologue).

RESPONSE: The reviewer is correct in that there are limited *Caenorhabditis elegans* orthologs identified and the GO terms are not listed in the table. Unfortunately, this is one of the challenges of working on a non-model organism - many genes simply don't have

orthologs, or they have one-to-many or many-to-many orthologous relationships. We have listed one-to-one orthologs.

Regarding providing the GO terms. We don't feel adding GO terms to the table would provide any useful resource for the reader; many genes will have none, some will have one, and some genes will have many. The Supplementary Tables 3, 4, 5, and 6 provide the genesets for each analysis, which can be used with gProfiler to replicate the analyses. Hence, it is the gene identifiers, and not the GO terms, which are useful to the reader.

To address this comment, we have expanded the methods to illustrate where the GO terms are found, ie. WormBase ParaSite. We have removed the *C. elegans* orthologs, and instead, provided InterProScan gene description and accessions from UniProtKB/TrEMBL hits to each gene in the Supplementary Tables 3, 4, 5, and 6, which does provide additional information about gene identification and putative functions of many genes that did not have a *C. elegans* ortholog in the original version.

R2.26: Anthelmintic resistance: The analysis again didn't identify anything specific in this study (probably because sampling was opportune not directed). There is something odd about the diversity of this locus in the ancient samples (not commented on in the text) (is this due to data processing? Creating a consensus would dampen diversity). Also what about the NHP samples? The baboon and Honduras samples might be a little different than China and Uganda but this would need statistical analysis.

RESPONSE: We chose to specifically look at beta-tubulin in our data as there are contentious results in the literature regarding whether this gene is involved in anthelmintic resistance. In veterinary nematodes, one of three key variants in beta-tubulin can mediate resistance to benzimidazole-class drugs. We don't have drug phenotype data for any of our samples, though it is likely that the modern human samples have been exposed given large scale mass drug administration would have been ongoing when the samples were collected, and that the adult samples were expelled from the gastrointestinal tract after drug treatment; ancient and baboon samples would not have been exposed to drugs. We hypothesised that drug resistance variants may be present in these populations, which might be a problem for future control. We found none of the previously described variants. We have remade the Supplement Figure showing the beta-tubulin gene model which hopefully makes this clearer.

We are not sure exactly what is odd about the ancient samples, other than across this large scaffold, the diversity is low. The vertical black line shows the position of beta-tubulin on the scaffold, and there is nothing obvious to describe here. No consensus was made.

Consistent with the comments above, we have chosen to remove the Tajima's D analyses. Instead, we now show the ranked distribution of Pi value throughout the scaffold to show that the beta-tubulin gene is not an outlier.

R2.27: Discussion>> The discussion often comments on the limitations of the data sets being analysed, which is appropriate but does call into question whether more data sets need to be analysed to make this study stand out and provide new insights above those already published with selected genetic markers (rather than whole genome data).

RESPONSE: We completely agree that more datasets would be beneficial. However, as argued above, this is impractical and not necessary to address the primary aims of this work.

Regarding "making this study stand out". This is the first population genomics analysis of a soil-transmitted helminth, and provides novel insights into the distribution of genetic diversity, both around the world and throughout its genome. It is one of the most comprehensive analyses of helminth genetic diversity to date. It includes analysis of ancient parasites, the oldest eukaryotic pathogens described using whole-genome sequencing to date. Our manuscript addresses questions regarding parasite adaptation, response to drugs, and fine-scale genetic variation, and makes a clear argument that whole genome-scale genetic variation is the appropriate resolution needed to address them. These points clearly stand apart from what has been published or what is possible with selected genetic markers.

Our discussion of the limitations is in fact a clear acknowledgement of where we believe future research could focus and build upon what we have done.

R2.28: Line 470 the comment that this data establishes a framework for genetic analysis should be toned down and indicate that it is a starting point.

RESPONSE: In the concluding paragraph, we describe this work using "establish" and "a significant first step", and both words allude to this being a starting point towards a bigger goal of genomic epidemiology. We do believe this work provides a framework, the basic essential structure, for developing genomic epidemiology for soil-transmitted helminths. We provide open access data and code to reproduce analyses without restriction, demonstrate how these data can and should be used, and identify key questions for future work. Anecdotally, within days of releasing the bioRxiv preprint of this manuscript, our genome resources were used by researchers in the US to detect and characterise soil-transmitted helminths. These data will continue to be developed and expanded upon by us and others to improve control of this important human pathogen.

R2.29: Data availability: The comment in materials and methods for the calling of 10 of the ancient sample mitochondria stimulate me to examine the PlosOne paper. Unfortunately in this there was no link to the accession files for this data. It is good to see that the raw read data is going to be available (will this be for all sequence data including those that didn't make it into the analysis?). It is important that the assemblies created as part of the current study also be made available.

RESPONSE: All raw sequence data generated for this study has been submitted to ENA and will be available for use without restriction.

To address this comment, we have amended the data availability statement to state that the assembled mitochondrial genomes are available for download from the Github repository, which will receive a stable DOI from Zenodo upon resubmission.

Reviewer #3 (Remarks to the Author):

This paper reports the first population genomic analysis of the important human parasite *Trichuris trichiura*, including samples that span the entire globe and the entire past millennium. Despite various technical challenges, including irrevocably pooled samples and low coverage for some samples, this is one of the most comprehensive population genomic surveys of a human helminth to date. These results will facilitate future efforts to understand and control this parasite, including monitoring and predicting how worm populations respond to drugs.

RESPONSE: We thank the reviewer for their very positive appraisal.

R3.1: The samples vary widely with respect to collection methodology and quality of the data. The details of these differences are largely explained, although it's not always clear which samples originate from the same patient, a crucial detail since worms from the same host are more likely to be closely related and may not reflect the full diversity of their population. The China samples all come from the same patient, and at least some of the Uganda samples come from the same patient, but not all? It would be good to note all of this explicitly. Samples can be classified as pooled, single worm low coverage, and single worm high coverage, but it's not obvious from the main text which worms are which; can this be indicated more clearly?

RESPONSE: The reviewer does make a good point, in that we were not very explicit as to how the worms might have been related to each other (i.e. from their sampling), other than to show that we tested for kinship between worms and only found only a single pair of samples that were likely related that were also sampled from the same host. The remaining pairwise relationships were weak and spurious and were not associated with within-host relationships.

We feel it would be difficult to be explicit about both discrete and continuous variables in a figure or text (without making it too confusing), hence why we have openly provided all of this information in the Supplementary data. As described above in response to another reviewer's comment, all of the modern worms were single worms, apart from the samples from Cameroon and Tanzania which we do state didn't sequence very well. The ancient samples are all pooled given they were eggs isolated from the environment.

To address this comment, we have:

- updated Supplementary table 1 with the relationship between worms based on host, in a new column called "SOURCE_DETAIL";
- updated the text to highlight single and pooled worms within each cohort;
- Generated a new Supplementary Figure 7 showing kinship between worms. Note these are only for samples that had sufficient nuclear genome coverage.

R3.2: Because of the differences in sample quality, there are caveats to analyzing all samples jointly. For example, in PCA, differences in apparent heterozygosity owing to differences in coverage could drive the results. It would be good to generate a PCA using haploidized data (drawing one allele at random from all sites in each sample) to test for this effect. Similarly, using pooled data probably violates the assumptions of NGSadmix, though the results appear to be robust to this since the ancient samples still form a distinct group along with the baboon samples from the same continent.

RESPONSE: This is a really good point, and something we had not explored fully. In response to this comment, we have generated three figures below that represent the same three plots in Figure 1: (i) mtDNA variants, (ii) zoom of mtDNA variants on mixed cluster, and (iii) autosomal variants. The data in these new plots was generated using ANGSD; a single randomly selected allele at each variant position from either mitochondrial and autosomal BAM files was used to generate an IBS matrix, from which a multidimensional scaling analysis was performed. These data faithfully recreate the broad-scale clustering seen using genotypes as presented in Figure 1. These data, together with the new analysis comparing coverage vs heterozygosity in Supplementary Figure 3, give us confidence that despite the

heterogeneity between samples, biological rather than technical differences are driving the patterns in genetic diversity we describe.

R3.3: Sex chromosomes and autosomes have different demographic histories and ideally should be analyzed separately. It's not explicitly stated which scaffolds in the figures (e.g. Fig 3) are sex-linked, and it would be good to clarify this. Also, genome-wide analyzes like PCA and summary statistics like π and Tajima's D should be done separately for autosomes and sex chromosomes. This can reveal sex differences in mating success or migration, and also autosomes may be more subject to phenomena like introgression from other species. An obvious thing to test is whether π on X-linked scaffolds is 75% of π on autosomes, which should be the null hypothesis. Why are there no genome-wide depictions of π , as there are for F_{st} (Fig 3) and Tajima's D (Supp Fig 11)?

RESPONSE: The review makes a really good point - we had that information, but didn't clearly show it.

In response to this comment, we have:

- Separated sex-linked and autosomal variants based on their linkage groups. The sex linked scaffolds belong to scaffolds containing "Trichuris_trichiura_1", whereas autosomal variants belong to scaffolds containing "Trichuris_trichiura_2" and "Trichuris_trichiura_3". Note that there are unplaced, small scaffolds that have not been assigned to a linkage group. As they contain a mix of sex-linked (including y-linked) and autosomal scaffolds, and many are collapsed repeats, we have not included them.
- We explicitly use only the autosomal variants in the PCA in Figure 1d, and admixture and population demographic analyses in Figure 3
- We use both sex-linked and autosomal variants in the genome-wide analyses in Figure 4. We now also include density plots to show the relative differences between populations, and between autosomes and sex chromosomes.
- We did determine the nucleotide diversity (π) of both sex-linked and autosomal scaffolds, of which the ratio is 0.72, which fits the null hypothesis of 0.75 as the reviewer suggests based on an XX/XY sex determination system.

- We have also now included genome-wide comparisons of P_i and D_{xy} , presented in a consistent way to the F_{st} data in the main text.

R3.4: A PCA of mitochondrial data (Fig1 b & c) is unusual and violates the assumptions of PCA since all mtDNA variants are in linkage disequilibrium with each other and are not independent. The mitochondrial phylogeny (Supp Fig 6b) is much clearer and evolutionarily valid. This, or a simplified version of it, should be in the main text instead of the mtDNA PCA.

RESPONSE: This is an interesting comment regarding the validity of using PCA to analyse mtDNA variation; there is certainly precedence in the literature for doing so, but there doesn't seem to be obvious arguments against doing so. We certainly agree that mtDNA variants are in linkage disequilibrium and are not independent. However, PCA doesn't assume (or require) data points with a sample are independent and can in fact be used via the loadings calculated to determine which data points are correlated and drive variance in each PC. PCA is broadly a "model-free" approach, at least compared with a phylogenetic tree, and is not used to infer anything more than broad-scale genetic diversity derived from correlated variant frequencies. In this context, it works as intended, to demonstrate geographic structure among populations, in which samples from the same population cluster together and are distinct from other samples in different populations.

We argue that the reason for performing the analysis in the first place and for keeping it in the main text is that there are a greater proportion of samples with sufficient coverage to analyse, including all of the ancient samples. This provides a nice contrast against the following PCA using nuclear variants (of which only 2 of 19 ancient samples are included); we gain additional resolution using more SNPs, at the cost of removing some samples due to lower coverage.

R3.5: Tajima's D is not biologically meaningful with fewer than four sequences, since all allele frequencies are identical. The authors note the high levels of positive Tajima's D on the two X chromosomes from baboon-derived worms; in fact Tajima's D is effectively impossible to calculate in this scenario and shouldn't even be mentioned or included in Supp Fig 11b.

RESPONSE: We acknowledge that Tajima's D has limited biological meaning in low input samples; in fact, we presented it simply to show variation across the genome, which does show discrete regions of population-specific differences. However, given we have not specifically used Tajima's D , we have decided to remove these analyses in line with the reviewer's recommendation.

R3.6: It would be nice to see mean heterozygosity estimates for each sample, perhaps in a supplemental table or figure. The decrease in π from Uganda to China to Honduras is one of the more salient results, and it would be good to know how robust it is. Do all Uganda samples, considered individually, have higher heterozygosity than all Honduras samples, for example? Heterozygosity may be confounded with coverage, so maybe plot heterozygosity versus coverage in order to show any patterns.

RESPONSE: The question regarding heterozygosity per individual per population is a good one, especially when considering that differences in coverage (especially low coverage) may bias this estimate.

In response to this comment, we determined heterozygosity per individual per population and compared it with nuclear coverage of the genome. For all modern populations, there were no significant correlations between heterozygosity and coverage (see new Supplementary Figure 3 in the manuscript). This suggests that differences in heterozygosity are not biased by differences in coverage and thus, our conclusion of differences in nucleotide diversity between populations still holds true.

There is, however, an interaction between heterozygosity and coverage in the ancient samples. These samples were pools and very low coverage, both of which may lead to an underestimation of variation present. In the nuclear analyses in the paper, only two of the 17 samples with the highest coverage are used that met the thresholds and thus, the overall impact of this bias is minimised.

R3.7: There is a lot of literature comparing the relative utility of F_{ST} and D_{xy} . Since opinions differ on these statistics, and it's clear that they can reveal different evolutionary phenomena, can D_{xy} also be shown?

RESPONSE: This is a good suggestion, given they are frequently used individually or together, and sometimes do show distinct patterns.

We now include D_{xy} in the supplementary data as Supplementary Figure 11, configured in the same format as the F_{ST} data. Arguably, it doesn't add anything obviously new or distinct from the F_{ST} data, and we find (perhaps not surprisingly) that it is reasonably correlated with the nucleotide diversity data per population. As such, we don't feel there is anything particularly noteworthy to include the data in the main text - drafts of this figure to show

both F_{ST} and D_{xy} side-by-side looked particularly noisy and reduced the ability to compare across samples.

R3.8: In Figure 1, is it possible to use the same symbols and colors in part A and in the later parts? For both the map and the PCA, it would be nice to see at a glance which samples are from which country and which ones are from human or other host species, etc., and also which are pooled.

RESPONSE: This is a good suggestion, as it was inconsistent. In response to this comment, we have updated the colour/shape scheme to be more consistent between the map and the PCAs. We have also continued this colour/shape scheme throughout the manuscript for consistency.

We have chosen not to show which samples are pooled vs single worm - we already have colors for location, shape for sample type, and there is no convenient way to add a third variable. We have however improved the text to say this more explicitly.

R3.9: In Fig 2b, are there 2 or 3 migration edges? There are three colored lines but the caption says two.

RESPONSE: Our apologies, this is inconsistent.

To address this comment, we have updated the figure legend to reflect there are three migration edges.

R3.10: Supp Figure 12a: Do the blue bars represent individual SNPs? If so, "Pi" is an unusual way to report them, since pi usually refers to segments more than 1bp long. Can this be stated as expected heterozygosity or minor allele frequency?

RESPONSE: Pi can be calculated per site (single SNP) or in windows (more than one SNP or position). Most tools commonly used to calculate Pi from genomic data can report both, eg. vcfTools used here does (--site-pi or --window-pi). This is because Pi is defined as the average number of nucleotide differences "per site" between two different sequences in all possible pairs in the sample populations. Given this analysis is focused on individual variants, ie putative benzimidazole resistance SNPs reported in the literature, it needs to show single base-pair resolution.

In response to this comment, we have updated the figure to improve the visualisation of the differences in both nucleotide diversity and minor allele frequency between variants, including the canonical resistant variants (now red and labelled).

REVIEWERS' COMMENTS

Reviewer #1 (Remarks to the Author):

The authors have addressed all my previous concerns and I am happy with the manuscript "as is".

Reviewer #2 (Remarks to the Author):

The authors have responded effectively to the questions raised in my review and made appropriate changes to the manuscript/figures (or provided good reasons not to do so in their rebuttal). No further changes are requested but the authors might wish to consider the following.

With reference to the age-related changes in diversity (and estimated population size) of the ancient samples (Fig2) it is entirely possible that Viborg is simply an outlier (either as a site or by virtue of containing greater damage associated diversity). Without this site there is unlikely to be an age related change in population estimates, perhaps the authors could consider briefly explaining this to the reader. Also if considering the populations of parasites it might be worth pointing out to the reader that the prevalence of infection in medieval Europe was as high as that seen in modern endemic areas.

Reviewer #3 (Remarks to the Author):

The authors have addressed my concerns.

Response to referees

Reviewer #1 (Remarks to the Author):

The authors have addressed all my previous concerns and I am happy with the manuscript "as is".

and

Reviewer #3 (Remarks to the Author):

The authors have addressed my concerns.

RESPONSE: We thank reviewers 1 and 3 for their time and positive appraisal of the revised manuscript, and that no further work is required.

Reviewer #2 (Remarks to the Author):

The authors have responded effectively to the questions raised in my review and made appropriate changes to the manuscript/figures (or provided good reasons not to do so in their rebuttal). No further changes are requested but the authors might wish to consider the following.

RESPONSE: We are glad to hear that review 2 was happy with our efforts to revise the manuscript, and believe the manuscript is better for it.

With reference to the age-related changes in diversity (and estimated population size) of the ancient samples (Fig2) it is entirely possible that Viborg is simply an outlier (either as a site or by virtue of containing greater damage associated diversity). Without this site there is unlikely to be an age related change in population estimates, perhaps the authors could consider briefly explaining this to the reader. Also if considering the populations of parasites it might be worth pointing out to the reader that the prevalence of infection in medieval Europe was as high as that seen in modern endemic areas.

RESPONSE: The reviewer is correct in that the Viborg samples could potentially be an outlier; we have no other samples of a similar age to test this hypothesis and removing these samples does reduce this association. However, we don't believe it is simply due to greater DNA damage, which is only low frequency and doesn't associate well with age in this sample dataset. Reanalysis of the same data after C-to-T and G-to-A substitution variants (that are susceptible to damage-associated variation) does show that this association holds true. Separately, we observe a similar pattern of lower nucleotide diversity and evidence of a contracting population size in the two ancient samples for which we have nuclear genome-wide data (AN_DNK_COG_EN_0012 & AN_NLD_KAM_EN_0034), neither of which were from Viborg. Therefore, we believe the data overall is consistent with age-related changes in diversity that we describe.

In response to this comment, we have updated the text in the results as follows:

“The oldest and genetically distinct VIB samples were also among the most diverse; this relationship between diversity and time was robust to DNA damage artefacts (deamination susceptible C-to-T and G-to-A substitution variants removed; $r=-0.49$, $p=0.047$), however, the trend was still negative but less pronounced and not significant when the VIB samples were removed ($r=-0.28$, $p=0.305$).”

Regarding the second comment, “it might be worth pointing out to the reader that the prevalence of infection in medieval Europe was as high as that seen in modern endemic areas”. We do in fact already describe this in the discussion:

“Epidemiological estimates of *T. trichiura* prevalence have been proposed to be comparable between medieval populations in Europe (dated between 680 and 1700 CE) and modern, non-European endemic regions”.

And so we don't feel repeating this statement would add anything new or helpful to the reader. However, to be fair, these estimates are based only a few sites in Europe (southern UK, and southern Germany and Czech Republic) and may not be representative of conditions at more northern latitudes, as we go on to discuss.